# Allocentric flocking

Mohammad Salahshour ◉ [1,2,3] ✉ & Iain D. Couzin ◉ [1,2,3]

Understanding how group-level dynamics arise from individual interactions remains a core challenge in collective behavior research. Traditional models assume animals follow simple behavioral rules, like explicitly aligning with neighbors. We present here an alternative theoretical framework that considers collective behavior to be grounded in neurobiological principles—particularly that animals employ ring attractor networks to encode bearings towards objects in space in an allocentric (i.e., with respect to a fixed external reference frame, such as a stable landmark) and/or egocentric (i.e., the angle relative to the animal's heading) neural coding. We find collective motion can emerge spontaneously when individuals act as sensory inputs to each other's networks, but only if individuals employ allocentric bearings to neighbors. Rapid switching between both representations can, however, enhance coordination. Collective motion can, therefore, emerge directly from navigational circuits, and thus may readily evolve, without requiring explicit alignment, or additional rules of interaction.

How collective behavior arises from interactions among individuals is central to multiple scientific disciplines[1–4]. A particularly notable example is collective motion; beyond its esthetic appeal, collective motion has been a testing ground for theories of collective behavior[5]. This is because the emergent macroscopic patterns arise from feedback between the individuals and the collective[6,7]. Traditionally, models of collective movement were rooted in agents following simple behavioral rules. Such studies have shown that emergent patterns can arise among such cognitively minimalist agents, termed 'self-propelled particles'. While the earliest such models included explicit alignment—such as the influential Vicsek model[8]—other models have shown that collective motion can arise from mechanisms like escape and pursuit[9], inelastic collisions[10,11], attractive and repulsive radial forces[12–18], active elastic forces[19,20], and nematic collisions[21], all of which can induce local alignment.

While suitable for inanimate objects or simple organisms, these modeling frameworks overlook the cognitive processes that shape individuals' perception of their physical and social environment[22–28]. This realization has led to more recent models that incorporate mechanisms like visual sensing of neighbors[22,29–31], and the explicit consideration of the sensory-motor interface, such as by incorporating biologically plausible mechanisms by which individuals may modify both their movements and their internal model of the world[32–36].

However, the vast majority of models still rely on traditional rule-based frameworks[12–14,37], such as self-propelled particles that employ explicit local alignment[31–33], thus divorcing the model from neural principles and experimental data[38–40].

Further to this, irrespective of their differences, all previous models of collective motion make a universal assumption: that vectorial information regarding conspecifics (the estimated directions/bearings towards others) is considered exclusively from an egocentric perspective. That is, it has always been assumed that, with respect to conspecific bearings, the frame of reference for a focal individual is with respect to its own, present, heading (for example, a neighbor positioned directly to the right would be at +90°, whereas one at the left would be considered −90°, with respect to the focal individual's heading, irrespective of its absolute heading (see Fig. 1). By contrast, however, neurobiological data demonstrate that the bearing towards external goals, even in simple animals such as the fruit fly (*Drosophila* species), can also be encoded in an allocentric (i.e., world-centered, such as north, south, east, west) frame of reference[41–43], and that such a representation is ubiquitous among animals[41–53].

Here, we propose a shift towards a modeling framework that accounts for the fact that animals are not rule-bound, self-propelled particles. Rather, we consider that they form neural representations of their environment and act on what they perceive. This perspective

[1]Department of Collective Behaviour, Max Planck Institute of Animal Behavior, Konstanz, Germany. [2]Centre for the Advanced Study of Collective Behaviour, University of Konstanz, Konstanz, Germany. [3]Department of Biology, University of Konstanz, Konstanz, Germany. ✉e-mail: salahshour.mohammad@gmail.com

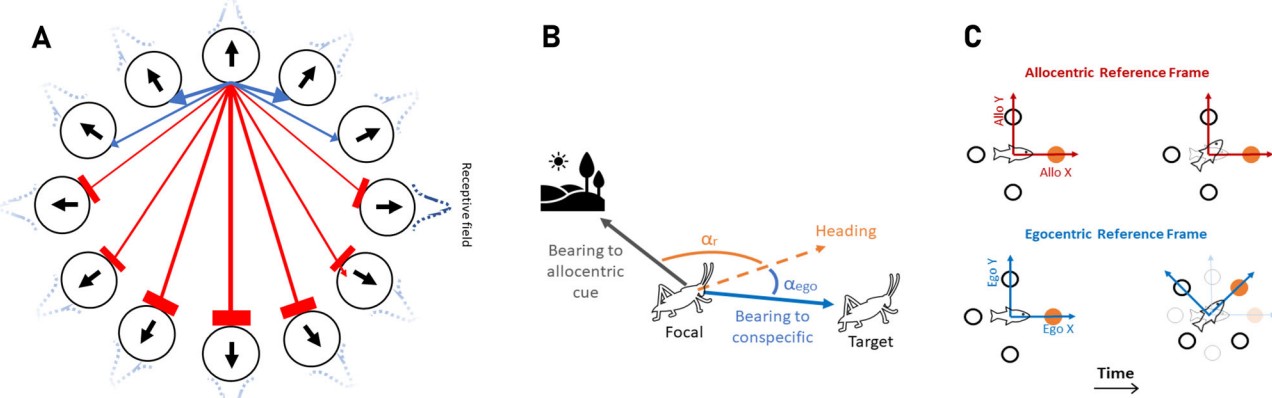

**Fig. 1 | Ring-attractor networks with an allocentric and an egocentric representation of space. A** Individuals are equipped with a ring-attractor network in which neurons are arranged on a ring. Each neuron receives sensory input from the external world through a Gaussian receptive field centered on an angle $\alpha_i$ (with respect to the individual's allocentric or egocentric reference frame) and encodes for movement along the same direction, $\alpha_i$. Besides, neurons interact with other neurons via excitatory or inhibitory synapses, depending on their distance along the ring. **B** With an egocentric representation of space, the animal encodes directions with respect to a self-body coordinate (head direction), $\alpha_{ego}$. Whereas, with an allocentric representation, directions towards targets are encoded via an allocentric frame of reference, $\alpha_{allo} = \alpha_r + \alpha_{ego}$, where $\alpha_r$ is the direction of the individual's body axis. Thus, with an allocentric representation, directions are independent of the agent's body coordinate. **C** To model an allocentric representation of space, we assume the neural network (represented by only four circles for better visibility) encodes for directions in a world-centric reference frame, which does not rotate with the individual's movement in space (as if it is anchored in the external world, using one or more external cues), such that, neuron $i$, encodes for a direction, $2\pi(i-1)/N_s$ with respect to a world-centric axis. In the egocentric case, the network's reference frame is attached to the individual and rotates with the individual as the individual moves in space, such that neuron $i$ encodes for a direction, $2\pi(i-1)/N_s$, with respect to the animal's body axis.

immediately opens the question of how animals' egocentric and/or allocentric representation of space affects collective behavior. We develop generative models of animal movement and decision making to address how both egocentric and allocentric representations may impact the establishment of goal vectors (the direction towards which the animal moves[41,43,44,53]), and thus movement decisions in individual and collective contexts.

We begin by considering how the internal dynamics of a ring-attractor network can give rise to spontaneous patterns of activity in the absence of any sensory input (e.g., during exploration/search), and the corresponding movement patterns that result. Following this, we explore how the neural dynamics, and thus movements, are influenced by simple sensory inputs. Specifically, we examine target-seeking behavior, both in response to static and moving targets in egocentric and allocentric representations. This relates the collective dynamics of the ring-attractor to information processing.

Finally, we consider the emergence of collective motion in such cognitive agents; here, the individuals are themselves salient sensory inputs to each other's ring-attractor network. Thus, the sensory input for each individual becomes much more complex, due to the geometric input to the ring depending both on self-generated motion and the movements of others. In doing so, we will demonstrate how collective motion can emerge directly from sensory information being integrated on a ring-attractor network. Notably, we find that an allocentric representation of space naturally results in the ability of animals to form coherent, mobile groups that exhibit a rich set of patterns. Although a purely egocentric encoding fails to produce collective motion, rapid alternations between allocentric and egocentric frames can enhance global order. The rich behaviors evident in what we term allocentric flocking, as well as the natural emergence of collective motion from neurobiological principles, call for a shift in perspective and a new class of models in the study of collective motion.

## Results
### The modeling framework
**Neurobiological motivations.** While neurobiological details differ among species, a ubiquitous motif for encoding angular information in both the invertebrate[54,55] and vertebrate[44,56–59] brain, are ring-attractor networks. A ring-attractor network is a recurrent neural circuit in which localized excitation and long-range inhibition maintain a bump of electrical activity, with recurrent excitation maintaining the bump even in the absence of sensory input. Ring attractors can have multiple inputs, often from other ring attractors and/or from sensory modalities. Their functional ring-like topology (which in some cases, such as the ellipsoid body of the fruit fly *Drosophila*, is literally also a morphological ring[54]), makes them ideal structures for the integration and representation of angular information.

Multiple interconnected and intercommunicating ring-attractor networks coexist in the brain. Central to spatial navigation is animals' neural compass, often termed their heading compass or head compass, in which the cellular activity rotates as the animal changes heading, allowing estimation of body/head orientation relative to visual[60] (and in some species also magnetic[61–63]) cues. Prominent visual cues employed to tether the compass include polarized light[64–66], the sun[60,61] and prominent distant, and therefore relatively stationary, objects in the environment[61,62]. In this way, the animal can maintain an allocentric reference frame for its heading, i.e., its orientation with respect to external cues[43,48,55,67–72]. While some species, such as fish[56], have rigid bodies, in others, such as mammals, animals can maintain head direction in addition to heading[57,58]. We note that head and heading directions have yet to be disambiguated in insects[54,55].

Maintaining a compass does not imply that each individual knows which way is north, or that different individuals share a common allocentric frame of reference; indeed, animals must typically re-tether their compass as they move through space and contemporaneous salient cues, e.g., visual[55] or magnetic cues[61,62,73], change. It only means that individuals can use available sensory information to maintain egocentric bearings, such as towards objects, as well as (thanks to their compass) to have the capacity to transform egocentric representations to allocentric representations on their ring-attractor networks[41,48,74]. Therefore, while all bearings we consider here are egocentric in terms of their point of origin—centered on the animal—their bearings can be encoded in an egocentric and/or an allocentric (polar) reference frame in the brain[48,55] (see Fig. 1). Importantly, this does not imply the

existence of a cognitive map or absolute knowledge of object locations in Cartesian space (e.g., knowing a tree's coordinates as (X, Y) independently of the individual's location[75,76]). Rather, we refer to the encoding of bearings in egocentric or allocentric terms[48]. Future work could extend this framework to incorporate more complex spatial representations, such as the Cartesian encoding observed in mammalian brains[50,75,76], but here we focus on the simpler mechanisms that may underpin the evolutionary origins of collective motion in invertebrates and vertebrates.

In addition to their heading compass, animals have also been found to encode their "goal direction" in a ring-attractor network. Here, the bump of activity represents the desired direction of travel for the animal at the present moment in time[41,43,44,53]. While it is known that animals turn towards their goal vector during navigation, with the neural circuitry responsible for converting allocentric goals into appropriate egocentric steering controls having been dissected in *Drosophila*[41,53], relatively little work has been conducted into how the goal vector is itself established when there are multiple alternatives[77].

Here, we focus our attention on this less-explored aspect of decision-making and make the reasonable assumption that animals can turn towards their goals. Thus, we do not explicitly model how animals maintain their allocentric heading, which they are known to be able to do, but rather how sensory information—with a specific focus on visual information—may be integrated to create time-varying goal directions.

Our use of a ring-attractor network to explore decision-making with respect to establishing a goal direction is motivated by its neurobiological plausibility[43], and that we previously found that a ring-attractor model could accurately predict the time-varying directional movement decisions exhibited by individual fruit flies, locusts, and zebrafish, in scenarios involving two or more discrete static (fruit flies and locusts) and moving (fruit flies, locusts and zebrafish) options[40,78,79]. In this work, we were, however, unable to account for how collective motion emerges in animal groups. Importantly, similar to all previous models of collective behavior, in our previous ring-attractor models[78,79], we had assumed that animals employ an egocentric representation of space.

**Generative models of spatial decision-making.** We develop a modeling framework to mechanistically capture how individuals' neural coding of their goal bearing[41,44,53], and hence their movement (see above), is governed by both spontaneous neural dynamics, as well as the neural processing of sensory cues. In our framework (see Methods), individuals' movement decisions are governed by a ring-attractor network. The network receives sensory input from the outside world and employs an internal collective dynamic to come to a consensus regarding the directional goal of the animal for that moment in time. We consider how goals can be established both in the absence and the presence of sensory information.

The sensory input to the network is assumed to be topographically mapped (as, for example, are visual stimuli in *Drosophila*), such that a perceived cue (e.g., visual target) excites the appropriate position (angle) on the ring[54,55,80]. While the model is agnostic to the modality, in many animal groups, vision is the primary modality. Because here we are interested in modeling individual movement in two-dimensional physical space (as an important starting point which has been the focus of most past theoretical and empirical works[8]), we assume a neuron at a position $\frac{(i-1)}{2\pi}$ along the ring to receive excitatory input from a Gaussian receptive field centered on an angle $\frac{(i-1)}{2\pi}$ with respect to the origin (zero) of the agent's reference frame (see Fig. 1A). This corresponds to the input to the ring-attractor being excitatory, with the potential to induce bumps of activity corresponding to one or more perceived targets. Much has been suggested occurs by the mapping of the inputs from the optic lobe to the protocerebral bridge in the fruit fly[55,80–82], and computationally this can be thought of as each

object inducing an external field on the ring (In flies for instance, visual cues arrive via the anterior visual pathway, a strictly topographic chain from the optic-lobe medulla through the anterior optic tubercle and bulb into the ellipsoid body and protocerebral bridge, where ring neurons then map those cues onto compass circuits[80–82]).

The interaction dynamics on the ring are mediated via local excitation and long-range inhibition, which typically results in a single bump, the consensus direction (the goal direction) being established. This is then translated into movement (i.e., we assume that individuals can move in their desired direction of travel). Inspired by ring-attractor networks observed in both invertebrates[54,55] and vertebrates[44,56–59], we take the recurrent connections to be a (generalized) cosine-shaped synaptic connectivity (preserving the ring structure of the network). This recurrent connectivity facilitates local excitation and long-range inhibition on the ring.

As the animal moves in space, the geometry of the inputs to the ring changes (because the relative position to targets changes), which changes the consensus goal, and so on. The movement of the animals in this model thus arises via an embodied, recursive process[78]. If there is no such sensory input, the goal is determined entirely by the internal neural dynamics of the ring attractor. If there is, sensory input can contribute to the collective neural dynamic on the ring and the resulting desired direction of travel.

In addition to formulating a deliberately simple model framework, which more easily allows us to identify which specific features of the computation of interest contribute to movement decisions, we also need to ensure that our findings are robust. To do so, we create two variants of our model:

1) Spin system model. We employ a spin representation of neural dynamics, originally proposed by Hopfield[83] to model associative memory and later used in diverse contexts, such as modeling animal decision-making in the presence of conflicting preferences[78,79,84]. In addition to its rich history, formulating neural interactions in this way provides access to tools from statistical mechanics, and despite its apparent abstracted nature, there exists a direct path from empirical neural data to this formulation[85]. Besides, mathematical mappings between spin system formulation and neural field formulation have been argued[78].

2) Neural field model. In the neural field model, we employ Amari's classical approach[86], originally proposed to model pattern formation in neural fields and later employed to study a wide range of continuous attractors, from ring attractors in head direction systems[87], to orientation tuning in visual cortex[88], working memory[89,90], and grid cells[91]. In these models, the stable bump of activity encodes a continuous variable (e.g., head direction or stimulus orientation) that is maintained over time in the absence of ongoing input. Beyond spatial orientation, the Amari framework has been extended to explore various cognitive processes. For instance, models of spatial working memory exploit persistent bump attractors to explain how information can be maintained temporarily without external cues by sustaining localized activity patterns[89,90]. Continuous attractor dynamics have also been applied to grid cell networks in the medial entorhinal cortex, providing a neural basis for path integration and spatial navigation (e.g., ref. 91).

Because we are interested in gaining insight into a wide range of movement and decision-making scenarios, we investigate the dependence of our results on noise in neural dynamics (resulting from intrinsic, extrinsic, and network-level noise). Consequently, we parameterize both models with an inverse noise parameter, $\beta$, where small $\beta$ values represent noisier neural dynamics.

A central focus of our work is how the representation of space impacts individual and collective behavior. Therefore, we consider both egocentric and allocentric representations on the ring attractor. When an agent possesses an allocentric representation of space, neurons encode for an allocentric, world-centric direction independent of the agent's orientation (e.g., the agent's head/heading).

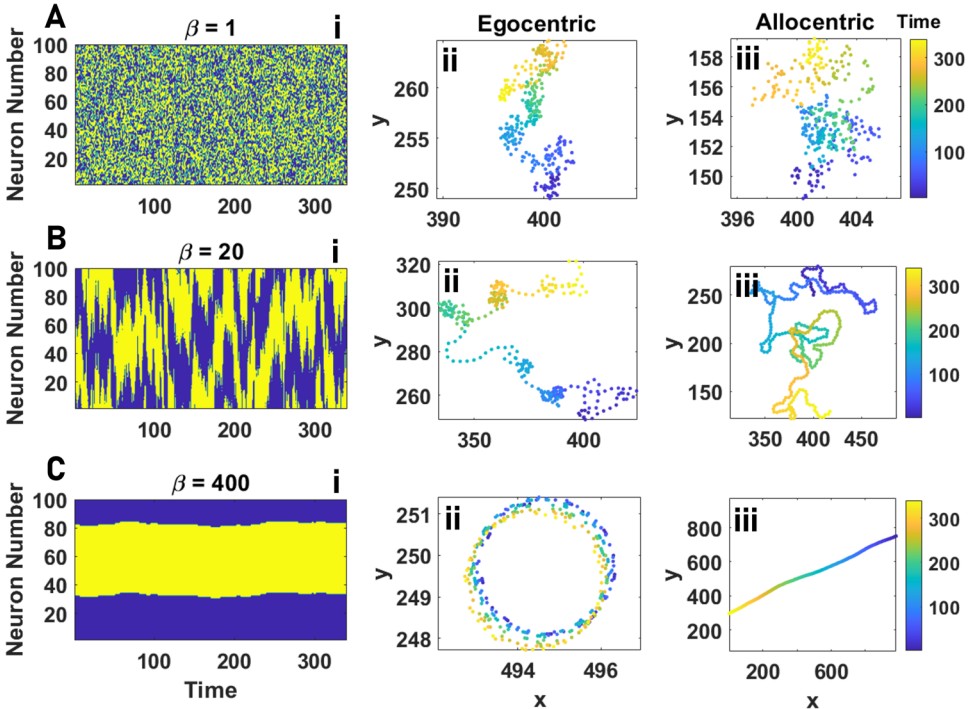

**Fig. 2 | Individual motion. A–C** The network activity as a function of time (**i**) and the resulting trajectories for egocentric **ii** and allocentric **iii** representation of space, for increasing values of $\beta$ from the disordered phase (small $\beta$, **A**) to the ordered phase (large $\beta$, **C**) are shown. In the disordered phase, the agent exhibits a random walk, and no difference between an allocentric and an egocentric representation of space is observed (**A**). As $\beta$ increases, differences become apparent. An egocentric representation of space results in a more meandering motion (**ii**), and an allocentric representation leads to a more directed motion (**iii**). For larger values of $\beta$, corresponding to the highly ordered network activity (**C**), motion patterns with an egocentric representation of space correspond to circular orbits (**ii**), and for an allocentric representation, correspond to a straight line (**iii**). Parameter values: $v_0 = 10$, $\sigma = 2\pi/N_s$, $h_0 = 0$, $h_b = 0$, $L = 1000$, and $N_s = 100$.

Biologically, this can be achieved, for instance, by the animal's ability to form an allocentric representation of its own heading, and thus, bearing towards an object via transformation of egocentric signals to allocentric signals[41,48]. Thus, while the agent's reference frame originates in the agent's position (i.e., it is attached to its body), it is anchored in the environmental cues and does not rotate as the agent moves in space and changes orientation. With an egocentric representation, on the other hand, objects are represented in an egocentric frame of reference (attached to the agent's body and rotating with its orientation). Thus, as the agent moves in space, its reference frame also rotates with the agent's body axis. See Fig. 1B, C, and Methods for details of model implementation.

We begin by presenting the results using our spin system model, and then present the neural field model. While some differences exist, suggesting that the two models can be complementary, both models predict key similar phenomenology. Most notably, both models predict that coherent collective motion can arise directly from animals' navigational circuits, with an allocentric, but not an egocentric representation of space.

### Individual motion and information acquisition in the spin system model

**Free individual motion.** First, we consider the intrinsic internal dynamics (i.e., spontaneous pattern formation on the ring structure) in the spin system model. We find that the system exhibits an order-disorder transition as a function of intrinsic (i.e., neural) noise, where the $\beta$ parameter is the inverse of noise; thus low $\beta$ corresponds to high noise, and vice versa; see the Supplementary Information S.1.

For values of $\beta$ below the critical transition point, the system is in the disordered phase, and there exists no correlated activity between adjacent locations on the network. Above the critical point, as $\beta$ increases (and thus noise decreases), order increases and correlations

emerge, but there is no stable (persistent) single bump of activity. For large values of $\beta$, however, we enter an ordered phase where adjacent spins (i.e., neural activity) assume similar states and a persistent bump of activity on the ring is observed. For details regarding the degeneracy of the ordered states and the dynamics at the critical point, see the Supplementary Information, S.1.).

How these intrinsic dynamics translate to animal movement depends on whether individuals maintain an egocentric or allocentric representation of space. This difference is not immediately evident for low values of $\beta$, such as below, or near, the critical point (Fig. 2A). This is because, in this regime, a bump of activity is highly unstable, resulting in agents' motion being slow and highly stochastic (similar to a random walk) for both egocentric and allocentric representations.

However, as $\beta$ increases, and the neural activity begins to exhibit a more stable bump (Fig. 2B(i)), different patterns of movement associated with each representation become evident at any spatial scale. For large $\beta$, but not too large such that the agent movement is still noisy, with an egocentric representation, the agent tends to often spend long times exploring small regions, with intermittent large jumps (Fig. 2B(ii)). Such a motion is not observed for an allocentric representation of space (Fig. 2B(iii)).

As $\beta$ increases further still (Fig. 2C(i)), the differences between the agent's motion with an egocentric and allocentric representation become most evident. For an egocentric representation, agents' motion tends to an imperfect circular trajectory (Fig. 2C(ii)), whereas for an allocentric representation, it is an imperfect directed path (Fig. 2C(iii)). Trajectories approach a perfect circular or directed path, respectively, only for the noiseless infinite $\beta$ limit. To make this comparison most directly, we illustrate how exactly the same neural dynamics result in very different types of motion: if neural activity is encoded in an egocentric way, a consistent bump position on the ring, $\alpha$ (corresponding to a deviation $\alpha$ of the neural bump from straight

ahead), requires the agent to constantly turn with respect to its direction, leading to a circular trajectory with radius $R = v_0/\alpha$ (if $\alpha = 0$, however, the agent will move in a straight path in its heading direction). By contrast, if the ring-attractor encodes angular information in an allocentric frame of reference, the position of the neural bump is independent of heading, leading to a directed trajectory along an angle $\alpha$ with respect to the agent's world coordinate system (e.g., an allocentric environmental cue).

**Response to an external sensory input—a target.** Now that we have an understanding of how the internal dynamics result in motion, we can investigate how an external sensory input to the ring-attractor network influences movement. We first consider the simplest case of a single, attractive, static, spatial 'target' (below, we extend this to consider the response to a mobile target). Even if the target itself is static, we can nonetheless consider this to be an information acquisition problem in a fluctuating environment because the agent's movement has the potential to continuously change the angular direction of sensory input, and thus the angular representation of the target. Thus, even the simplest form of target-seeking is an embodied process, where the motion of the individual may impact the geometric representation of the target, which, in turn, impacts the network activity and thus the resulting individual motion, and so on. This is compounded by the dynamics induced by the motion of the target with respect to the individual in the mobile target case.

We are interested in two aspects of target-seeking; (1) how quickly a target is found (i.e., how quickly an individual comes into close proximity to a target, following which it may employ a 'stopping rule'[92,93]), and (2) how well an individual can maintain close proximity over time (i.e., how well it can track a target). For both egocentric and allocentric representations, individuals move towards the targets. However, the representation employed results in differences between the types of trajectories exhibited. We begin by presenting movement patterns for a fixed target.

**Finding and staying close to a static target.** The network dynamics and the trajectory resulting from those dynamics for $\beta$ values ranging from the disordered phase to the ordered phase, when the agent faces a fixed target, are presented in Fig. 3. For high noise (small $\beta$), below the critical point, agents' movement is predominantly random. However, the external input on the network can induce weak selective movement towards the target. Consequently, agents tend to move slowly toward the target, with speed increasing as $\beta$ increases. In this regime, we do not observe differences between egocentric (Fig. 3A) and allocentric (Fig. 3B) representations since each agent's movement is predominantly random. See S.3 for details.

As $\beta$ exceeds the critical point, we begin to observe clear differences in the trajectories exhibited by egocentric and allocentric agents, with these differences becoming increasingly visible as we move towards the relatively high values of $\beta$ that characterize the ordered regime. Notably, if employing an egocentric representation, the neural input corresponding to the detection of the target tends to stabilize the bump, and individuals employ a meandering, but relatively direct, path toward the target, and stay confined to a small region when reaching a target (Fig. 3C for moderate and Fig. 3E for large values of $\beta$).

If employing an allocentric representation, by contrast, a bump of activity represents a specific bearing in the world. The presence of a target can destabilize this bump. Such a destabilizing effect is not observed for moderate values of $\beta$ (compare Fig. 3C and D). For larger values of $\beta$, due to the destabilization of the bump, instead of the smooth transitions in the network state that we observed in the absence of an external stimulus, we now find that the network dynamics show sudden transitions between different bumps. This results in a rich set of patterns of motion, such as inward spiraling

motion towards the target, corresponding to damped traveling waves of the bump on the ring-attractor network (Fig. 3F).

In addition to quantifying the trajectories, we can also ask how individuals may be able to optimize their ability to locate a target. For the fixed target scenario, the agent faces an easy task. In such a simple environment, successful information acquisition can be achieved by simply finding the target and then remaining close to it. We find that target seeking is optimized when the neural network is near the critical point (see S.3). While allocentric and egocentric representations of space perform equally well in such a simple task, if close to criticality, egocentric agents outperform allocentric ones in the ordered phase (see the S.3). This is due to the fact that agents with egocentric representation, once find the target, can stay close to the target by slowing down and/or settling on an attractor with a small radius. On the other hand, an allocentric representation of space can make such a simple task unnecessarily difficult, as allocentric agents need to constantly transition between their attractors (bump of activity) to maintain a bump of activity which accurately encodes the relative position of the target. This can lead to reduced performance.

While finding and remaining close to a target can be of importance in many contexts, in others, successful decision-making may require the agent to only find a target. For example, an animal may consume the target, or the animal may have a stopping rule that they employ once they reach the target[92,93]. Thus, it is important to address how fast the agent can find a target. To address how allocentric and egocentric perceptions of space affect decision-making time, in Fig. 3G, we present the decision time required for the agent to reach a static target. The results indicate that the decision-making time is optimized in the ordered phase. Furthermore, the agent's decision-making speed is higher (i.e., time taken is lower) with an allocentric representation of space. In the Supplementary Information we show that when higher accuracy in finding the target is required, that is, when successful decision-making requires the agent to reach a closer proximity of a target, an egocentric representation of space can become more advantageous for large values of $\beta$ (S.3).

**Tracking a moving target.** We move on to the problem of tracking a moving target. When the target speed is sufficiently small compared to the agent's average speed, the situation is similar to a fixed target. This is illustrated in Fig. 3H, where the average distance of the agent to the target is plotted. However, the situation changes in a fast-changing environment, where the agent needs to track a target with an appreciable speed relative to the agent's average speed. In Fig. 3I, we present the distance of the agent to a moving target with a high speed. While for a slowly moving target, egocentric agents can outperform allocentric ones, with a high target speed, the existence of an allocentric representation provides a benefit because, when employing an egocentric representation, the agent's movement can lead to dramatic fictitious changes in the external world, brought about by the shifting agent's position (we note that, this is not a problem when seeking a fixed target, because in such cases, the agent can use a simple strategy of standing still once finding the target). Accounting for these changes requires large changes in the network activity to constantly encode a mobile target's position. On the other hand, the environmental change resulting from the agent's movement is not dramatic from an allocentric perspective. Consequently, allocentric agents can easily modify their movement trajectory by small shifts in their bumps of activity.

We also find that, while higher values of $\beta$ decreases time to reach the target (which can be considered decision-making speed), the ability of the agent to stay in close proximity to the target (which can be considered decision-making accuracy) is higher for lower values of $\beta$. This trade-off is effectively solved not at the critical point, but in the ordered phase. This is due to the fact that following a moving target requires a more coherent movement of the agent, which can only be achieved in the ordered phase. Furthermore, the distance of the

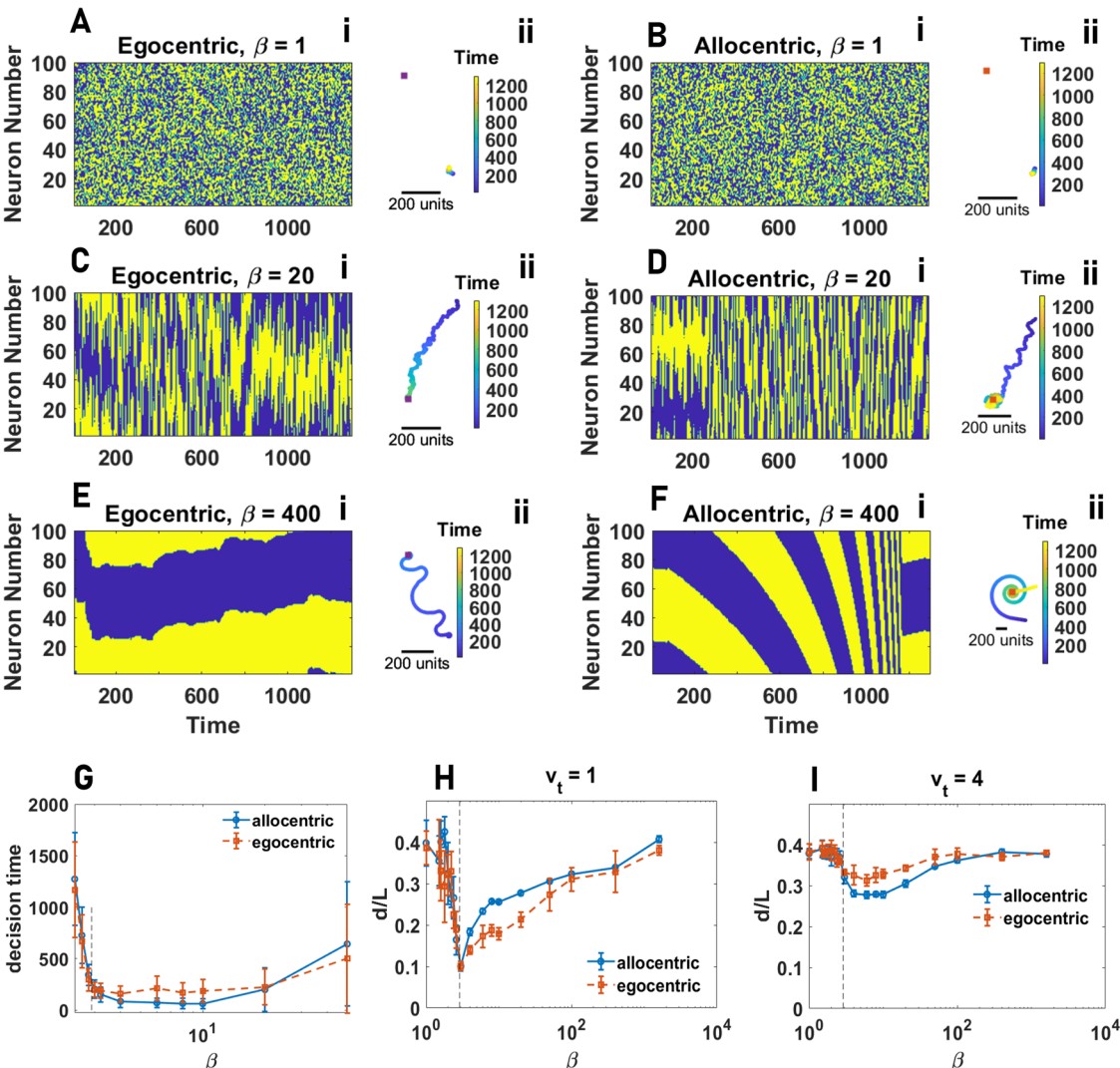

**Fig. 3 | Individual information acquisition. A–F** The network activity as a function of time (**i**) and the resulting trajectories (**ii**) for egocentric and allocentric representations for increasing values of $\beta$ is shown. For too small $\beta$, for both egocentric and allocentric representations of space, the agent only exhibits random and slow movement. Above the order-disorder transition, the agent moves towards the target. For smaller values of $\beta$, noise drives transitions between states, which facilitate information acquisition by endowing the agent with flexibility. In the ordered phase, external stimuli elicit distinct network activities for allocentric and egocentric representations of space. With an allocentric representation, external stimuli can lead to the formation of damped traveling waves corresponding to spiral motion toward the target (with more stability for larger values of $\beta$). For too large $\beta$, trajectories intermittently veer away from the target. With an egocentric representation, external stimuli help stabilize a bump of activity, allowing agents to remain stationary once it has found the target. **G** The agent's decision-making time

in finding a stationary target, defined as the time needed for the agent to reach close proximity of the target (5 dimensionless units), is plotted as a function of $\beta$. Decision-making time is minimized in the ordered phase. An allocentric representation can improve the decision-making speed in the effective decision-making region. **H, I** The time average distance of the agent to the target, normalized by the arena size $L = 1000$, $d/L$, as a function of $\beta$ for both allocentric and egocentric representations and for different target speeds, is plotted. For a stationary or slowly moving target, an egocentric representation is beneficial by allowing the agent to stay stationary once it finds the target. However, for larger target speeds, an allocentric representation improves information acquisition by facilitating the tracking of a rapidly moving target. In both cases, the information acquisition optimizes in the ordered phase but is close to the critical point. Parameter values: $v_0 = 10$, $\sigma = 2\pi/N_s$, $L = 1000$, $h_0 = 0.0025$, and $h_b = 0$. **A–F** $N_s = 100$, and in **G–I** $N_s = 400$.

effective decision-making region to criticality increases as the target's speed increases.

## Individual motion and information acquisition in the neural field model

While the phenomenology of the neural field model is generally similar to the spin system model, differences are nonetheless observed in their dynamics. The network in the neural field model does not exhibit an order-disorder (critical) transition, which is a characteristic of spin systems in statistical physics[94]. Rather, as $\beta$ increases, the network transitions from a state where the membrane potential of all the neurons is close to zero to a state where a bump of activity is sustained in

the network (see S.10). Consequently, the agent transitions from an immobile state to exhibiting directed movement, and a random walk-like behavior is not observed for any value of $\beta$. Both models are, however, comparable in the regime where a single bump is maintained (high $\beta$). Namely, with an allocentric representation of space, directed trajectories are observed, and an egocentric representation of space leads to circular trajectories with varying radii, including a directed trajectory (i.e., infinite radii). See S.10 for details.

When considering tracking a moving or stationary target, we observe a generally similar phenomenology to the spin system model. Namely, the information acquisition capacity of the agent is maximized for intermediate values of $\beta$, where the agent exhibits more

flexible decision-making. For too small values of $\beta$, the agent does not move or moves too slowly. For too large values of $\beta$, the agent lacks flexibility and performs poorly in finding or tracking a target. An allocentric agent performs better than an egocentric agent in tracking a moving target in the neural field model. In finding and staying close to a fixed target, allocentric and egocentric agents perform equally well for small $\beta$ (this contrasts with the spin system model, in which egocentric agents perform better than allocentric ones for such values of $\beta$), but allocentric agents outperform an egocentric one for large $\beta$ (similarly to the spin system model). See S.10 for details.

In the presence of a target, the neural field model also differs in several ways from the spin system model. For example, a stationary agent in the absence of a target can start to move (or a moving agent can increase its speed) towards the target, when the target is introduced. In addition, the threshold $\beta$ value above which the agent starts to move is shifted to lower values in the presence of target(s) in the neural field model. Furthermore, in the egocentric neural field model (unlike the spin system model), we observe spiral-like trajectories during which the agent slowly moves towards, or away from, the target. See S.10 for details.

## Collective motion in the spin system model

Above, we demonstrated that even for simple sensory inputs, having either an egocentric or allocentric neural representation in the ring-attractor network can greatly impact movement. Now we consider the far more complex sensory environment experienced by individuals in social groups and ask how egocentric and allocentric representations of bearings towards others impact collective movement. Here, the individuals themselves become static, or mobile, targets from the perspective of others. Thus, the neural ring-attractor dynamics of each individual both influences and is influenced by that of others, as a recursive (recurrent) feedback loop.

The model of collective movement straightforwardly results from the individual movement model by having a population of $N$ agents, each of which is a target for others, with an amplitude of the receptive field, $h_0^s$. Collective movement can thus be studied using a single parameter, a social attraction parameter, $h_0^s$, which parametrizes the strength of social attraction (see Methods). As the control parameter of the model, we consider the total social attraction, $h_t^s$, defined as the social attraction of an individual towards another individual, $h_0^s$, times the population size, $N$. As we will see, multiplying social attraction acting on an individual due to each other individual, $h_0^s$, by the population size results in a similar phase diagram for different population sizes. This is implemented by taking the control parameter of the model to be $h_t^s$.

In the main text, we focus on the global order and local order of the system. Global order (GO) is calculated as the sum of the normalized velocity vectors of all individuals (equivalent to the order parameter of the Vicsek model[8], and the alignment/polarization of the system as employed in collective behavior studies[5]). Local order (LO) is the average normalized velocity, not of the whole system, but of each individual's local topological neighborhood. This allows us to differentiate, for example, between disordered dynamics at all scales, such as when there is disorder, and thus low local or global alignment, and states where there is low global order, but high local order, such as when populations are composed of multiple small, coherently moving groups, but each tends to move in a different direction. See Methods for details and Supplementary Information, S.4–S.6 for the supplementary analysis (e.g., using measures of distance between agents).

## Egocentric representation of space

We begin by considering the emergence of collective motion for individuals that employ an egocentric representation. In Fig. 4A, B, we plot the global and local order parameters in the plane defined by $\beta$ and $h_t^s$. As the strength of social attraction among individuals increases, we see that populations cannot achieve global order (Fig. 4A), and thus, large-scale collective motion is never observed. However, we find that local order tends to be moderate to high (Fig. 4B). Thus, agents' direction of travel is similar to their close neighbors, but this emergent alignment is highly localized. At the scale of the population, increasing social attraction results in the formation of aggregation (but not collective motion), where agents coalesce in a dense group with low mean nearest neighbor and all pair distance (see S.5).

The distribution of GO and LO is presented in Fig. 5A, B. While the lack of global order is manifest in the insensitivity of GO to variation of social attraction (Fig. 5A), LO shows bimodality close to the phase transition (Fig. 5B). This shows that this phase transition is discontinuous (see S.5 for details). A high value of local order in the ordered phase indicates that agents' heading points in a similar direction compared to their neighbors. However, this local order does not induce collective motion. Instead, agents form a compact group. A snapshot of the agents in this phase is presented in Fig. 5C. For smaller values of $\beta$, presented in Fig. 5D, a similar situation is observed, however, in this case, noise in individual agents' movement drives a random walk-like motion within the cohesive aggregation of agents.

## Allocentric representation of space

The situation changes dramatically if agents possess an allocentric representation. Now three distinct regimes are observed. This can be seen in Fig. 4C, D, where the GO and LO as a function of $\beta$ and $h_t^s$ are plotted. As social attraction increases, the system shows a phase transition from a disordered phase to a phase where both local and global order are high, indicating the emergence of large-scale collective motion out of agents' inclination to stay with the group. Further increasing $h_t^s$, yet another phase transition from the collective motion phase to an aggregation phase where agents form a relatively immobile aggregation with high LO and low GO is observed.

For relatively high neural noise (low $\beta$), large-scale collective motion is never observed, as we only see a cross-over from the disordered phase to the aggregation phase where agents form a compact pack in which local order is observed, but no collective motion emerges (Fig. 4C). This is because, in our model, collective motion is a collective information acquisition problem and emerges due to agents coming to a consensus regarding their direction of travel. Thus, coherent large-scale collective motion requires agents to be able to keep a spatially-consistent bump of neural activity over time, which can only occur if neural dynamics are not too noisy (i.e., are in the ordered phase, Fig. 2F). We term this allocentric flocking.

## Allocentric flocking and population (system) size

By studying the statistical properties of the different regimes of allocentric flocking, we find that the motion patterns observed are sensitive to system size.

For very small system sizes, such as $N = 10$, the order-disorder transition (the transition from disordered motion to collective motion) exhibits bistability. This can be seen in Fig. 6A, where the distribution of global order close to the disorder-order transition shows two peaks, one at low and the other at high, GO (representing the disordered and ordered phase, respectively). This indicates intermittency between disordered motion and ordered motion. We find that agents show a wide range of collective motion patterns including swirling, sudden expansions (similar to flash expansions exhibited by animal groups[95–97]), fission-fusion dynamics, as well as coherent, directed motion. Figure 6E presents a snapshot of motion patterns during swirling when the group rotates around a common origin, and Fig. 6F for a swirling, resulting in a coil-shaped trajectory. See the Supplementary Videos 1 and 2. In this regime, global order is low, but local order is high, and the distance between agents exhibits strong fluctuations over time. This can be seen in the blue line in Fig. 6C, D, where GO and the mean distance between all pairs for different values of $h_t^s$ are plotted.

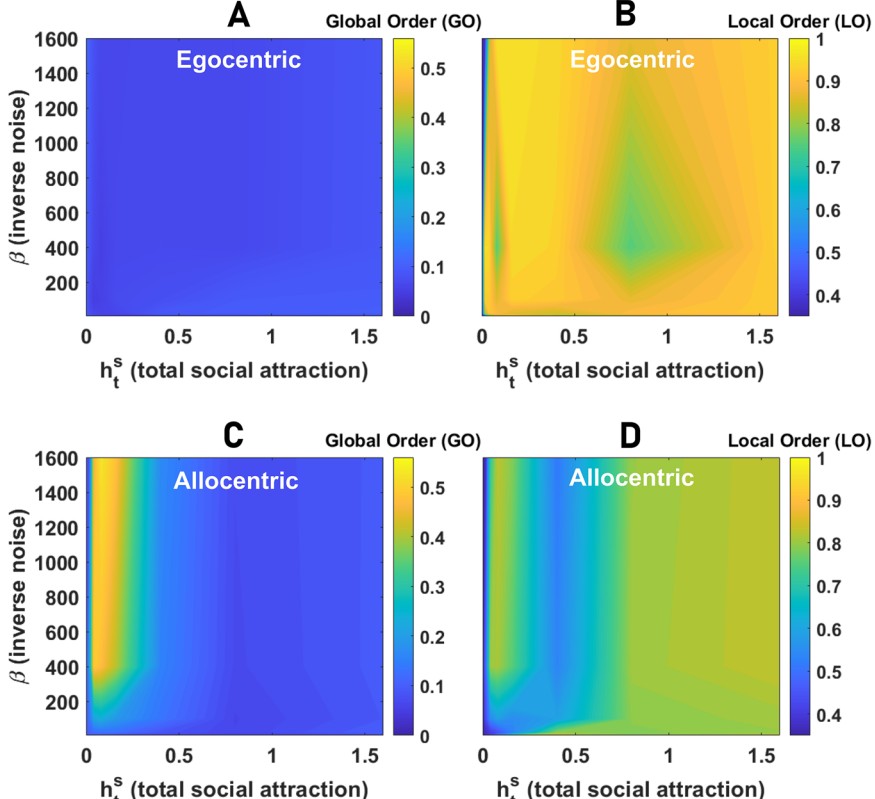

**Fig. 4 | Collective behavior of agents with egocentric and allocentric representation of space. A, B** Global order (GO in **A**), defined as the angular order parameter (AOP), and Local Order (LO in **B**), defined as the topological vectorial order parameter (VOP), in groups of 80 agents with an egocentric representation of space are color plotted as a function of the network inverse temperature, $\beta$, and total social attraction, $h_t^s$. For a too small social attraction, the agents move independently. As the social attraction increases, local order increases but not global order, indicating the onset of an aggregation phase where agents aggregate in a stationary dense group. **C, D** The global (**C**) and local (**D**) order in groups of 80 agents with allocentric representation of space as a function of the network inverse temperature, $\beta$, and total social attraction, $h_t^s$, are color plotted. The system shows three distinct phases: disordered motion for small $h_t^s$, collective motion with high local and global order, and aggregation phase with low global but high local order. Local order is minimized close to the phase transition between collective motion and aggregation due to the strong fission-fusion dynamics leading to explosive movement of the densely packed group. Parameter values: $N_s = 100$, $v_0 = 10$, $\sigma = 2\pi/N_s$, $h_b = 0$, $N = 80$, and $L = 1000$.

In Fig. 6G, we present a snapshot of the motion patterns for directed motion in relatively small groups ($N = 10$, see Supplementary Video SV.3). In this example, agents form a coherent, mobile group (as exemplified in Fig. 6C, D, red line). This trajectory also shows an example of intermittency between such directed motion and the formation of a stationary aggregation (where global order transiently decreases (orange line in Fig. 6C, D), resulting in a (probabilistically likely) reorientation once the group transitions back to coherent motion. This aspect of collective behavior is reflected in the bimodal distribution of local and global order parameters, as shown in Fig. 6B (see the S.4 for details).

As population size increases, however, the situation changes. While transitions in collective state appear discontinuously in small system sizes, they become more continuous for larger system sizes. This is shown in Fig. 7A, B for the disordered-collective motion, and collective motion to aggregation phase transitions, respectively (Here $N = 320$. See S.5 for other system sizes). In larger populations, intermittency between different motion patterns is more frequent (Fig. 7C, D). Furthermore, due to the prominent fission-fusion dynamics, the population is more likely to be decomposed into different groups exhibiting different collective motion patterns, such as collective motion, swirling, explosive movement, or sudden direction changes. This results in global order being a less effective means of characterizing the collective behaviors exhibited by agents. Examples of some motion patterns, including collective motion and fission-fusion

dynamics, are presented in Fig. 7E, F. See the Supplementary Videos 4, 5, 6 and 7. Notably, to reach different motion patterns, it is not necessary to tune the parameters of the model; Rather, diverse motion patterns occur for the same parameter values and are exhibited by the same population of individuals over time.

By increasing the social attraction, the system shows a continuous phase transition to an aggregation phase where the population forms a dense group of agents lacking directed motion. However, near the transition, frequent explosive and implosive movements of the population are observed (this results in a decline in local order in the phase transition region, as observed in Fig. 4D). A snapshot of the motion pattern in this phase is plotted in Fig. 7G. See the Supplementary Video 8.

**Cognitive representation during collective movement.** In our model, collective motion results from simple feedback between the ring-attractor networks employed for spatial navigation by animals. Key to the patterns observed in the collective context, such as sudden and coordinated changes in direction of mobile groups, is the synchronization of the neural dynamics of the agents. In Fig. 8, we present a snapshot of collective motion in a population of 80 agents. The population can be decomposed into subgroups of coherently moving agents whose neural dynamics exhibit synchronization. Evaluating the spatio-temporal dynamics of the neural representation in the brain of each agent, we see there exist emergent leader-follower dynamics, as

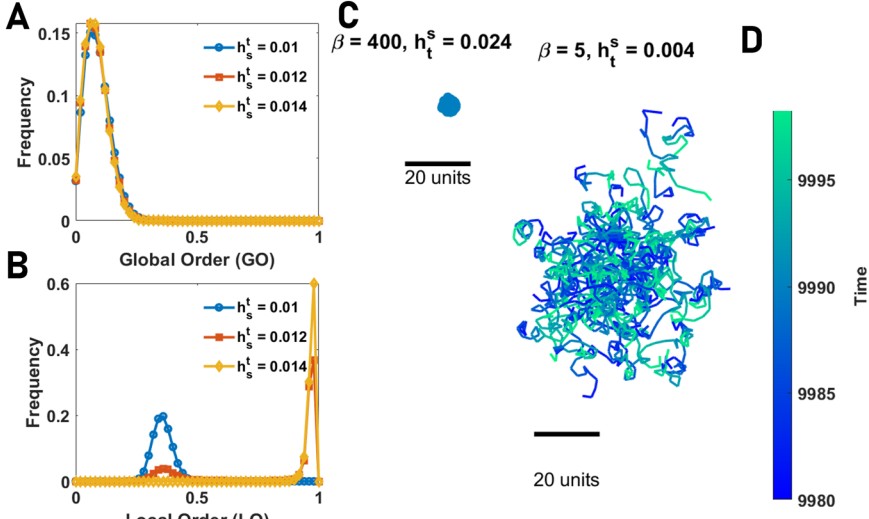

**Fig. 5 | Phase transitions in collectives with an egocentric representation of space. A, B** The distributions of global order (GO) and local order (LO) in groups of various sizes of agents with egocentric representation of space for different values of total social attraction, $h_t^s$, are plotted. By increasing $h_t^s$, the system shows a phase transition from a disordered phase, where individuals move independently, to an aggregation phase, where individuals form aggregates, and no collective motion is observed. While GO takes a small value and does not show sensitivity to social attraction (indicating no collective motion exists), LO shows bimodality close to the order-disorder transition, indicating a discontinuous transition from the disordered phase with low LO to the aggregation phase with high LO. **C, D** Snapshots of the collective behavior in the ordered phase are shown. For large $\beta$ (low network noise), in the ordered phase, agents form an almost stationary circular pack of densely aggregated agents. For smaller $\beta$, the pack's radius increases, and agents perform a random walk-like movement within the pack. **C** shows a snapshot of a dense pack for large $\beta$ and **D** shows trajectories of the individuals within a pack for smaller values of $\beta$. Local order is high in both cases. Parameter values: $N_s = 100$, $v_0 = 10$, $\sigma = 2\pi/N_s$, $h_b = 0$, $\beta = 400$, $N = 80$, and $L = 1000$.

evident by direction changes of an individual (change in the position of the neural activity bump on their ring-attractor network) being followed by similar changes in others (Fig. 8B–C) or fission-fusion dynamics, whereby an individual (or subgroups) desynchronize (resulting in fission) or synchronize (resulting in fusion) with others (Fig. 8D, E). See S.7 for more details.

## Collective motion in the neural field model

Because the spin system model and the neural field model have different formulations, including in the implementation of neural dynamics, and coupling of the neural dynamics and spatial decisions, a direct comparison of the two models is not straightforward. For instance, while we use the same label for $\beta$ as an inverse noise parameter, the implementation of noise in the two models is different. Nevertheless, in the context of collective behavior, we observe key similarities, but also some differences. In Fig. 9 we present global and local order in a population of 80 agents, whose decision-making is governed by the neural field model. Collective motion is not observed with an egocentric representation of space. In this case, as the social attraction increases, the collective forms an aggregation. In contrast to the spin system model, however, in the aggregation phase, high local order is not observed, indicating that the group remains in a disordered state in this case (see S.11 for details). With an allocentric representation of bearings, on the other hand, as social attraction increases, a phase transition to collective motion with relatively high global and local order is observed (see S.11). Further increasing social attraction leads to a second phase transition where the population collapses into a nearly stationary aggregation, with a relatively high local order but a low global order (see S.11).

Differences between the two models are also observed in the collective movement patterns. While some of the motion patterns exhibited by the neural field model are similar to those exhibited by the spin system model (see SV.9 for $N = 80$ and SV.13 for $N = 320$), additional patterns that are not, or only weakly, observed in the spin system model emerge in the neural field model. An example of motion patterns, where the collective, exhibit milling (SV.11), or transitions between the milling and directed motion can be observed in SV.10. In addition, close to the aggregation phase, the neural field model can also exhibit patterns of subgroups of aggregated, yet collectively moving individuals in relatively large groups (SV. 12 for $N = 80$ and SV.14 for $N = 320$).

## Switch between allocentric and egocentric representations of space

So far, we have considered an allocentric and an egocentric representation of space as separate cases. While such a scenario allows us to study what phenomena are associated with each mode of perception and decision-making, animals can maintain[41,45,46,48,49,98], or switch[99,100] between, allocentric and egocentric representations of bearings towards objects in space. To address how such a coexistence or switching between the two representations affects individuals and collective behavior, here we consider a simple scenario where agents can switch between an allocentric and an egocentric representation at random, such that, in each timestep, with probability $\omega$, the agent employs an egocentric representation and with probability $1-\omega$ it employs an allocentric representation of space. We assume that agents have the capacity to continuously map their allocentric and egocentric reference frames. This capacity is implemented in our model by taking the goal direction of the agent as the zero of the agent's reference frame, following a switch from egocentric to allocentric (or from allocentric to egocentric) representations. This re-anchoring ensures that switching between reference frames does not lead to the animal getting lost following a switch. The fact that allocentric and egocentric neural processing coexists in animals, suggests that animals have such a capacity, for instance, via information processing in different levels (brain regions)[45,47], path integration[48,50,52], or flexible use of allocentric and egocentric spatial memories[46]. See Methods for details.

In Fig. 10, we plot the global and local order in the $h_t^s - \omega$ plane. When switching between allocentric and egocentric representations of space is slow, the agent effectively uses either allocentric (small $\omega$) or

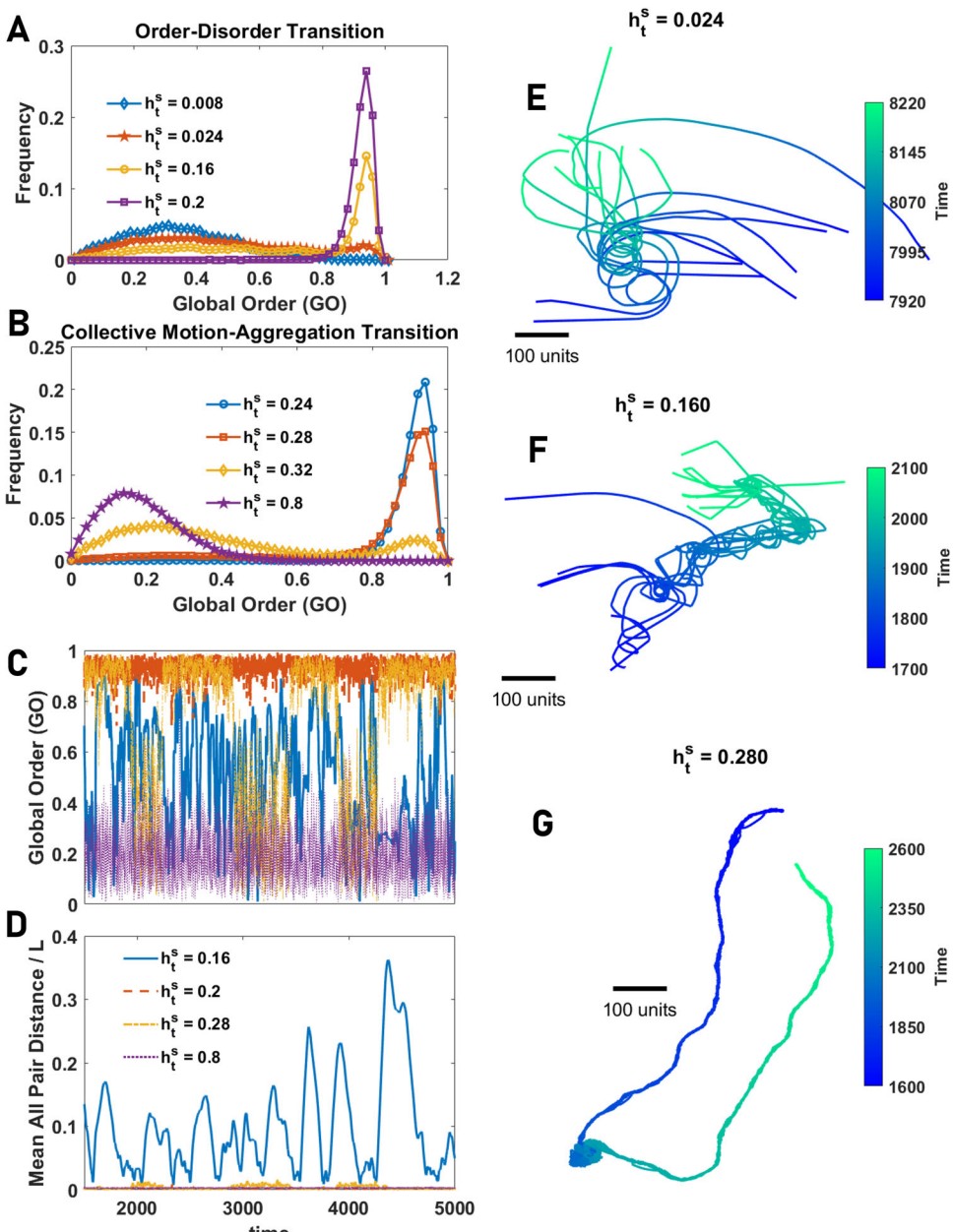

**Fig. 6 | Collective behavior in small groups of agents with an allocentric representation of space. A, B** The distribution of GO and LO in groups of various sizes of agents with allocentric representation of space close to the order-disorder transition (**A**) and close to the collective motion-aggregation phase transition (**B**). In both cases, the distribution shows bimodality, signaling a discontinuous pseudo-phase transition in small system sizes. **C, D** GO and mean distance between all pairs normalized by space size, $L$, as a function of time for different values of total social attraction are plotted. For small social attraction, the system shows intermittency between high and low order, and for larger social attraction, intermittency between ordered motion and aggregation is observed. **E–G** Example snapshots of motion patterns from the Supplementary Videos for some values of $h_t^s$ shown in (**A, B**). In the collective motion phase, the system shows a rich set of motion patterns, including swirling in circular orbits (**E**), or in coil-shaped orbits (**F**), fission-fusion dynamics, and intermittency between highly ordered motion and aggregation (**G**). Parameter values: $N_s = 100$, $v_0 = 10$, $\sigma = 2\pi/N_s$, $h_b = 0$, $\beta = 400$, $N = 10$, and $L = 1000$.

egocentric (large $\omega$) navigational strategies at each time. In these cases, we do not observe differences in motion patterns with cases where the agent employs purely allocentric or purely egocentric representations of space. That is, for small $\omega$, collective motion with high global and local order, similar to those observed for a purely allocentric navigational strategy, is observed, and for large $\omega$, collective motion is not observed.

On the other hand, when such switches are fast compared to the spatio-temporal scales of the agent's movement, such that the vectorial representation of bearings towards conspecifics does not undergo dramatic changes between switches, the random switch can effectively lead to the coexistence of the two reference frames, such that the agent's movement is the result of (temporal) integration of both allocentric and egocentric representations. In this region, we observe differences in motion patterns, manifested in two maximal regions for global order in Fig. 10, which occur close to the order-disorder (small $h_t^s$) and collective motion-aggregation (large $h_t^s$) phase transitions.

To gain insight into why this happens, in Fig. 10, we present the snapshots of motion patterns when the agent does not switch and employs a purely allocentric representation of space ($\omega = 0$) and when it switches at a rate which maximizes global order (compare SV.9 for

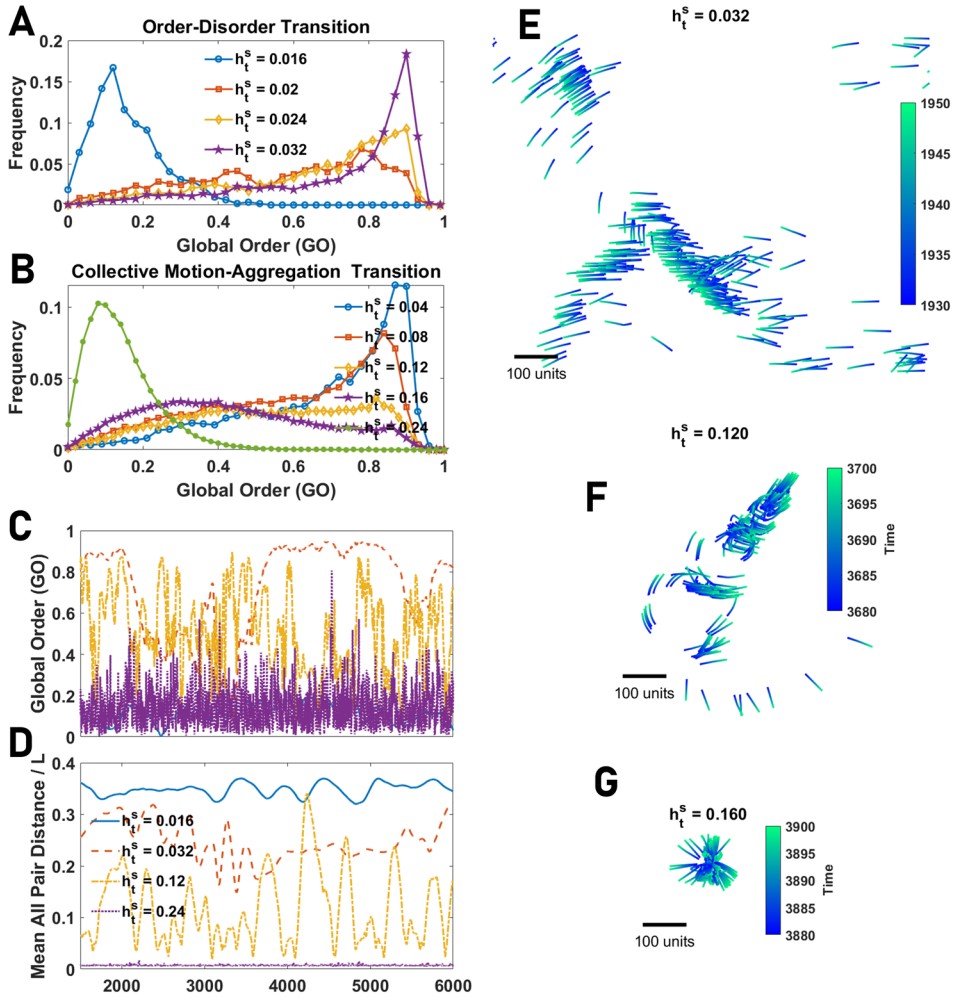

**Fig. 7 | Collective behavior in large groups of agents with allocentric representation of space. A, B** The distribution of GO (AOP) and LO (normalized topological VOP) in groups of various sizes of agents with an allocentric representation of space close to the order-disorder transition (**A**) and close to the collective motion-aggregation phase transition (**B**). Both phase transitions tend to a continuous phase transition, indicated by large fluctuations and a broad distribution of the order parameter, as group size increases. **C, D** Global order (GO) and mean distance between all pairs normalized by space size, $L$, as a function of time for different values of total social attraction are plotted. The system shows

intermittency between high and low orders resulting from transitions between different motion patterns and strong fission-fusion dynamics. **E–G** Example snapshots of motion patterns from the Supplementary Videos for some values of $h_t^s$ shown in **A, B**. In the collective motion phase, the system shows a rich set of motion patterns, including flocking (**E**), sudden direction change and fission-fusion dynamics (**F**), and intermittency between highly ordered motion and aggregation leading to explosive movement with low local order (**G**). Parameter values: $N_s = 100$, $v_0 = 10$, $h_b = 0$, $\sigma = 2\pi/N_s$, $\beta = 400$, $N = 320$, and $L = 1000$.

purely allocentric and SV.16 for the switching rate which maximizes global order with otherwise the same parameter values). These snapshots correspond to the maximal region at small $h_t^s$, near the order-disorder phase transition. As can be seen in SV.9 (see S.11 for the time series of collective motion metrics), employing an allocentric representation of space can decrease the stability of the flock, especially when individuals are too close to each other. While endowing the group with some of its esthetic characteristics observed in biological systems (such as sudden, coordinated direction changes, leading to relatively high fluctuations in collective motion metrics as a function of time, see S.11), this can be detrimental to global order. On the other hand, a random, fast switch between allocentric and egocentric perceptions of space simplifies the task of staying close, and moving together, and leads to highly ordered motion (see S.12). This is consistent with our previous finding that an egocentric representation of space simplifies the task of staying close to a stationary target (as in a coherently moving flock vectorial representation of conspecifics is subject to only relatively small or no temporal changes).

Switching between an allocentric and egocentric representation of bearings, at a rate maximizing global order, endows a similar benefit close to the collective motion-aggregation phase transition (the maximal GO region for large $h_t^s$ in Fig. 10), by stabilizing moving subgroups of aggregated individuals (see SV.16). We note that in both cases (both maximal GO regions in Fig. 10), similar patterns of highly ordered collective motion are also observed in a purely allocentric flock (as can be seen in SV.9 and SV.12). However, while such a simplified form of collective motion is often intermittently observed and can become unstable in purely allocentric flocks, it becomes highly stable, when collectives combine allocentric and egocentric navigational cues, leading to more stable bumps of activity in their ring-attractor network.

## Parameter dependence
In the Supplementary Information, using both spin system and neural field models, we confirm that the phenomenology of the model holds for other parameter values, such as the complexity of the agents (the

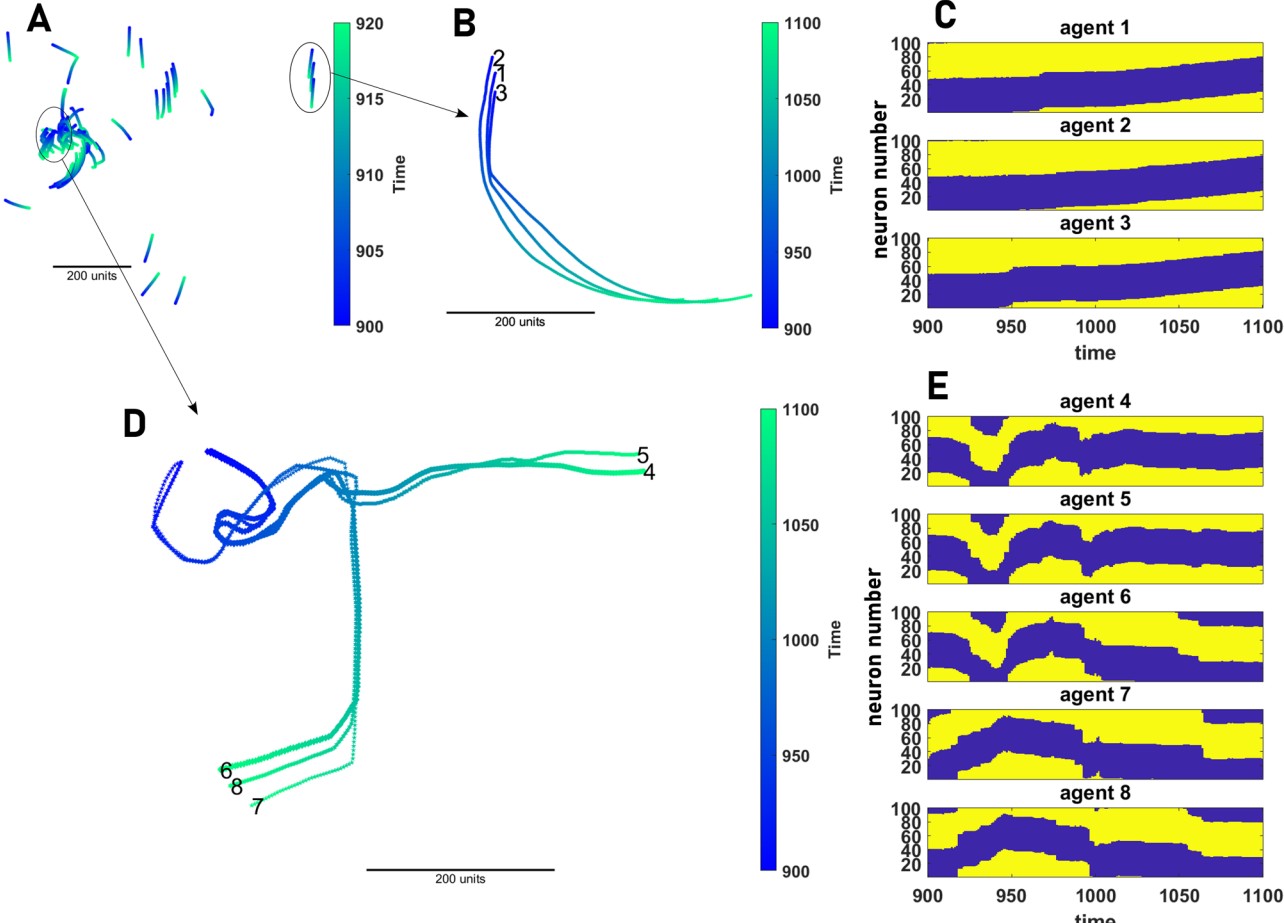

**Fig. 8 | Cognitive representation of collective motion. A** A snapshot of the collective motion in a population of 80 agents is shown. The population can be decomposed into subgroups of synchronized agents. **B** and **C** The motion pattern (**B**) and the neural activity (**C**) of a subgroups of three coherently moving agents among the 80 agents presented in (**A**) are shown. The coordinated movement of the agents results from the synchronization of their neural dynamics. **D**, **E** An example of fission-fusion and leader-follower dynamics is shown. In the beginning, agents 4–6 are synchronized and move together, and agents 7 and 8 move together. When these two groups come into close proximity, at around timestep 1000, agent 6 changes its mind and joins agents 6 and 7. Consequently, its neural activity becomes synchronized with agents 7 and 8. Around time 1050, a sudden direction change by agent 6 drives a sudden direction change in agent 8, followed by a similar behavior of agent 7. Parameter values: $N_s = 100$, $v_0 = 10$, $\sigma = 2\pi/N_s$, $h_b = 0$, $\beta = 400$, $N = 80$, and $L = 1000$.

number of neurons), speed constant, $v_0$, and the width of the receptive field of the agents (S.6 and S.11). For instance, when considering the effect of density, by increasing the size of the arena or decreasing the number of agents, we observe that in contrast to classical models based only on local alignment[8], density does not affect the phase transitions. Rather, similar phase transitions are observed at both low and high population densities, indicating a lack of density-dependent phase transitions. In contrast to classical self-propelled-particle models, which include only local alignment and therefore predict a density-dependent order-disorder phase transition, our model reproduces recent empirical evidence showing that no such transition occurs in locust swarms[40].

Besides, we show that our findings are robust when short-range repulsion is introduced to the model (S.8), or when social attraction decays with distance (S.9). Moreover, we consider a modification of the neural field model where agents move with constant speed. Their direction of motion, however, is determined by their neural network. We show that this modified model gives rise to similar phenomenology. In addition, we show that when recurrent connections are removed (by taking the synaptic connectivities between all neurons to be zero, $J_{ij} = 0$), agents show a rather trivial form of motion by collapsing into a relatively stationary aggregate (S.14). An example of such dynamics is provided in SV.17.

## Discussion

At the level of individual navigation, our models suggest that an egocentric representation of space may provide advantages in navigating in relatively stationary environments and towards nearby objects. These findings are consistent with empirical observations that some organisms tend to represent nearby objects more in an egocentric, and those far away more in an allocentric way[100]. On the other hand, our models predict that allocentric representations facilitate the pursuit of moving targets. This seems to be consistent with empirical findings in both insects[41] and mammals[44].

Our main finding, however, is that a rich suite of collective behaviors, including the formation of coherent, mobile groups, emerges naturally from the types of neural circuits—ring-attractor neural networks—employed by animals during spatial navigation. By contrast to classical models of collective motion that use hypothetical rule-based interactions, such as repulsion, alignment, and attraction, our modelling framework is grounded in cognitive principles of spatial information processing in the (invertebrate and vertebrate) brain. We show that collective motion can emerge directly from navigational circuits, without requiring explicit alignment, or additional rules of interaction —if individuals employ an allocentric (but not an egocentric) representation of space. While not previously considered in the study of collective behavior, this spatial representation is known to be

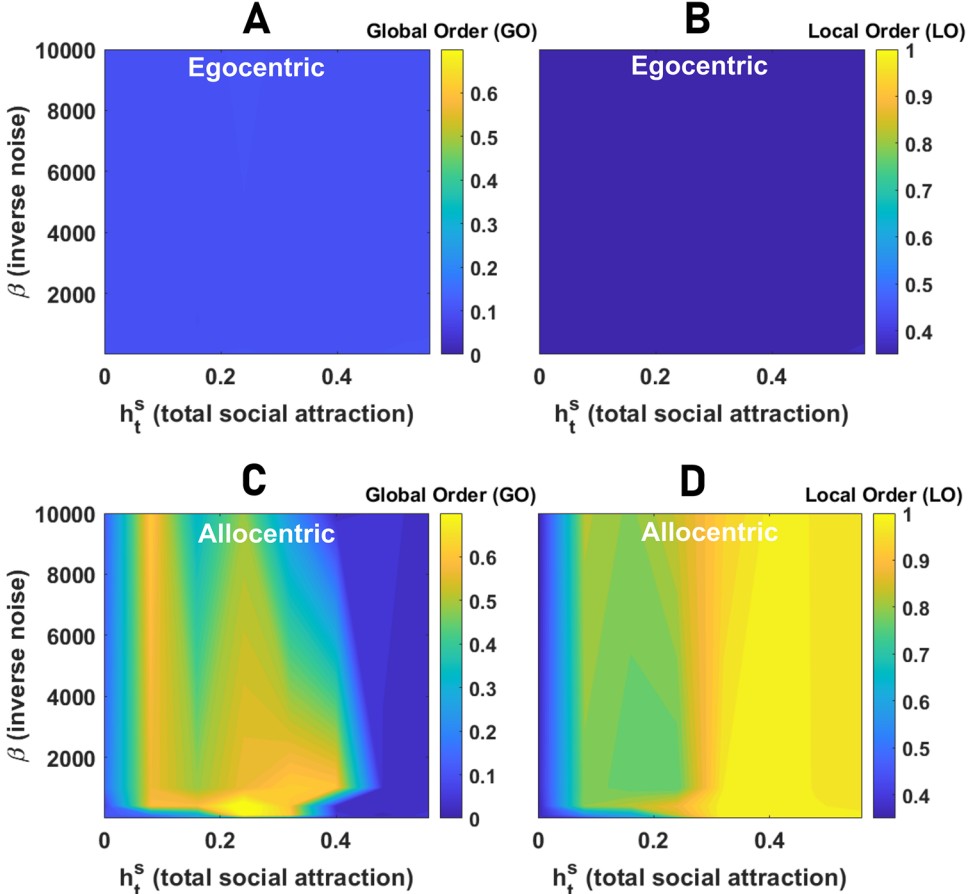

**Fig. 9 | Collective behavior in the neural field model. A, B** Global order (GO in **A**), defined as the angular order parameter (AOP), and Local Order (LO in **B**), defined as the topological vectorial order parameter (VOP), in groups of 80 agents with an egocentric representation of space are color plotted as a function of the network inverse temperature, $\beta$, and total social attraction, $h_t^s$. Both local and global order remain small, indicating collective movement is not observed with an egocentric representation of space. **C, D** The same quantities for an allocentric representation of space are plotted. Similarly to the spin system model, the system shows disordered motion for small $h_t^s$, collective motion with high local and global order, and aggregation phase with low global but high local order. Parameter values: $N_s = 100$, $v = 0.5$, $v_0 = 0.05$, $\sigma = 0.4$, $h_b = 0$, $N = 80$, $\Delta t = 0.3$, and $L = 1000$.

ubiquitous, employed by fruit flies[41], and humans[47], alike. Following the historical use of the term flocking to broadly describe the collective dynamics of diverse systems—whether physical particles, animals, or robots[5]—we term this mechanism allocentric flocking.

While alignment is observed in many species—such as starlings[101,102] and shoaling fish[38,39,103]—our work suggests it may be an emergent by-product of allocentric (or coexistence of allocentric and egocentric) representation of space by animals– a view supported by studies that failed to find empirical evidence for explicit alignment among fish[38,39] or swarm-forming locusts[40]. Our work shows that local alignment can arise as a form of consensus dynamic (not dissimilar, conceptually, to models of collective information acquisition[104]) for agents who have an allocentric representation of bearings (their own heading direction, and the bearings towards others, are within a world-centred frame). We show that, by contrast, if individuals exhibit an egocentric representation (whereby bearings are body-centred, but directional bearings are only encoded with reference to the present heading), social attraction can only result in the formation of relatively immobile aggregations. Here, the additive nature of attraction is analogous to gravitational collapse[105].

Allocentric flocking results from interactions among cognitive agents with an allocentric representation of space, where individuals themselves act as sensory inputs to each other's ring-attractor networks. While at an individual level, we find that an allocentric representation of space can be beneficial for effective target-seeking in a

rapidly changing environment, it is shown to be essential to achieve coherent collective motion. We also considered the fact that animals can employ both egocentric and allocentric representations of space, with the ability to integrate and/or transition between them (e.g. rapid resets to landmarks[46,48,50], temporal switching[99,100], or coexistence of both via parallel information processing in different brain regions[45,47]). Using a minimal random switching scheme, we find that rapid, intermittent flips between frames (with their attendant re-anchorings) can enhance global alignment beyond the pure-allocentric case. Whether, and if so, how animals schedule their frame switches, as well as implementing more sophisticated context-dependent switching and/or integration of such representations, is a promising avenue for future work.

Despite the differences in their mathematical formulations, both the spin system and neural field models arrive at the same core prediction: an allocentric encoding of bearings is essential for the emergence of coordinated motion. However, they also exhibit differences that suggest avenues for empirical testing. For instance, the spin system model predicts random walk-like motion for high neural noise (low $\beta$), a pattern not observed in the neural field model. Conversely, the neural field model predicts speed changes in the presence of targets when $\beta$ is small, with higher speeds occurring in response to more attractive stimuli. Furthermore, in the presence of a target, the egocentric neural field model can exhibit trajectory patterns distinct from those of the spin system model, such as spiral-like paths or slow

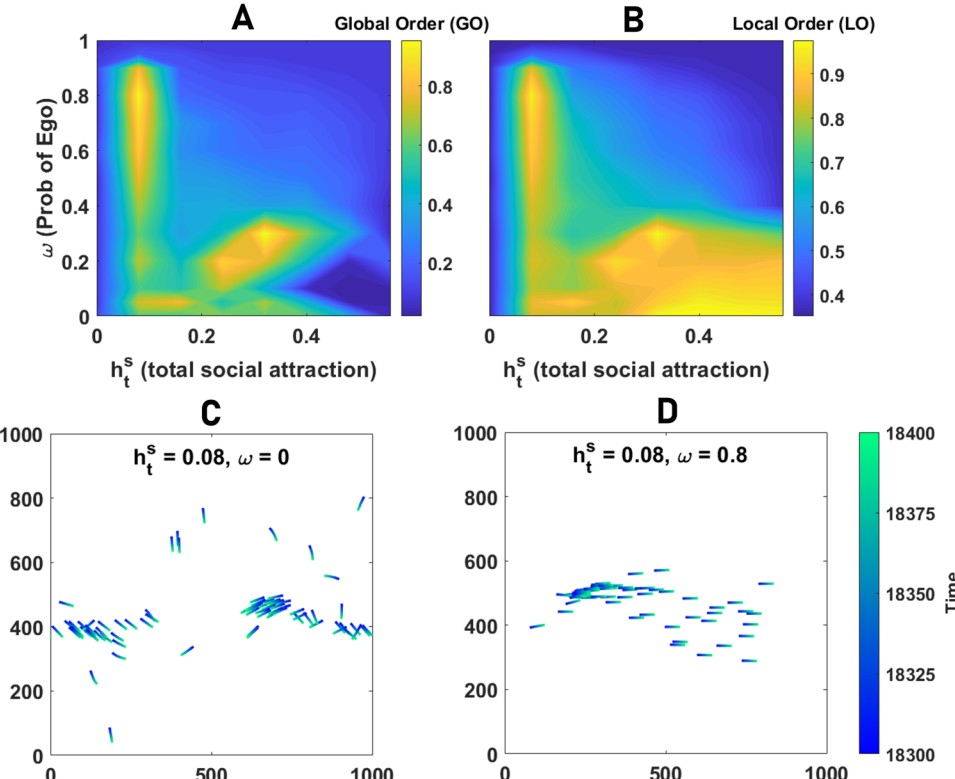

**Fig. 10 | Switching between allocentric and egocentric reference frames.** Global order (**A**) and local order (**B**) in the neural field model, where individuals randomly switch between allocentric and egocentric representations of space, are color plotted as a function of $h_t^s$ and the probability of being in the egocentric state, $\omega$. A certain rate of random switch between allocentric and egocentric representations can increase global and local order by stabilizing highly ordered collective motion at the expense of reduced complexity of the motion patterns. **C**, **D** show snapshots of collective motion close to the maximal order region when individuals possess a purely allocentric representation of space (**C**) and when they switch at a rate close to the rate leading to maximal order (**D**). Parameter values: $N_s = 100$, $v = 0.5$, $v_0 = 0.05$, $\sigma = 0.4$, $h_b = 0$, $N = 80$, $\Delta t = 0.3$, $\beta = 1000$, and $L = 1000$.

movement away from an attractive stimulus when in very close proximity. Considering population-level properties, while the spin system model predicts there can be local, but not global, alignment of headings with an egocentric representation, no local order emerges in the neural field model. In addition, while similarities in collective patterns exist, our results suggest that milling behavior can be observed in some parameter ranges in the neural field model. We have not observed this pattern in the spin system model.

Here, we have shown that collective motion, along with a diverse set of patterns observed in animal groups, can arise directly from animals' navigational circuits. While this provides a mechanistic explanation, it is important to note that collective motion[106,107] and its distinct patterns[106]—such as fission-fusion dynamics[108,109], swirling/milling[110], and flash expansion[95–97]—may offer functional advantages and, therefore, be favored and shaped by evolution under different ecological contexts. In this regard, our model suggests that, although these patterns may also serve functional and evolutionary purposes, they can equally have proximate causes (rather than, or in addition to, evolutionary ones). Incorporating evolutionary perspectives into our framework—for example, by allowing our cognitive agents to evolve under different ecological scenarios—could help address how different motion patterns evolve, potentially by driving the system into specific parameter regimes and shaping the agents' sensory-motor integration in different ecological contexts. Such functional considerations could clarify how evolution shapes animal navigational circuits and their perception of the environment to better meet ecological and environmental demands. Ultimately, this approach may help explain—both mechanistically and functionally—why and how collective motion and its diversity emerge in a wide range of biological populations.

In conclusion, allocentric flocking provides a contrasting, but empirically grounded, explanation as to how collective behavior may arise in many animal species. It demonstrates how easily collective behavior can emerge from known neurobiological principles and thus may readily evolve from an asocial ancestral state. This parsimonious framework can also be readily modified to be adapted to specific systems to incorporate further features, such as individual and collective learning, or to address different ecological questions, such as collective sensing, navigation, and decision-making. Besides, while our work is inspired by biological neural networks, it can potentially integrate biological and artificial neural networks and motivate new areas of research in artificial neural networks, such as swarm robotics. By introducing allocentric flocking as a general mechanism for the emergence of collective behavior, we hope to encourage further research into the feedback loop between neural dynamics and organismal collective behaviors.

## Methods
### The modeling framework
We employ two formulations of neural dynamics to provide parsimonious generative models of animal movement and perceptual decision-making. In both formulations, we provide simple models where a neural network is equipped with sensory input and provides motor output. We use spin system formulation of neural dynamics based on spin variables originally proposed by Hopfield[83], and Amari's neural field model[86], which formulates neural dynamics based on membrane potential, to formulate our models. In the Supplementary Information S.17, we generalize our neural field model by providing a conceptual framework based on a general $m$-layer network.

**The Spin System Model.** We consider cognitive agents capable of sensing and decision-making. The agent's decisions are governed by a ring-attractor neural network with $N_s$ neural groups (to which we sometimes refer as neurons or spins) and endowed with a ring structure. In the spin system model, neural groups are modeled as spin variables, following Hopfield formation of neural dynamics[83], and can take two states, active, $+1$, and inactive, $-1$. The activity of each neural group, $i$, is determined by an input from other neurons, $\sum_j J_{ij}\sigma_j$, and an external field, $h_i$, indicating the sensory input on the ring.

We parametrize neural groups by a discrete variable, $\alpha_i = \frac{2\pi(i-1)}{N_s}$, indicating their position on the ring. Without loss of generality, the angle $\alpha_i$ is measured with respect to the zero of the agent's reference frame, $O$, e.g., the agent's head. The external field on each neural group, $h_i$, is determined based on the sensory input the neural group receives. Each neural group, $i$, has a receptive field centered around the angle $\hat{\alpha}_i$. A neural group responds to external stimuli based on the angular deviation of the stimuli from its receptive field center. We will work with a Gaussian response function given by the following equation:

$$h_i = \frac{h_0}{\sqrt{2\pi\sigma^2}}\exp\left[-\frac{|\hat{\alpha}_i - \theta_{\text{target}}|^2}{2\sigma^2}\right] \tag{1}$$

The key question, where the agent's representation of space plays a role, is how to specify the angles, $\hat{\alpha}_i$. With an egocentric representation of space, the angles $\hat{\alpha}_i$ encode for a polar direction in the body-centered coordinate of the animal. Thus, we can take, $\hat{\alpha}_i = \alpha_i$. This is implemented by indexing neuron with respect to an arbitrary position of the animal, e.g., head, such that neuron $i=1$ with angle $\alpha_1 = 0$ represents the heading direction of the animal (assuming the animal's head is aligned with the direction that it is heading to), and neuron $i$ receives input from an angle centered on $\alpha_i = \frac{2\pi(i-1)}{N_s}$ with respect to the animal's heading. We can think of the animal as always turning towards the direction that it is moving to (heading direction), such that the animal instantaneously updates its head to coincide with its heading direction (see Fig. 1C). We note that this assumption originates from the parsimony of our model and its focus on addressing how animals establish a goal direction (rather than, e.g., focusing on how animals maintain an allocentric head direction with respect to landmarks[87]), making the reasonable assumption that they can steer towards their goal[41,44,53,77]. Furthermore, biologically, this intuition is supported by empirical data based on which head and heading direction coincide (in animals with rigid bodies such as fish[56]) or are strongly correlated[41,54,55,68,69]).

In the allocentric version of our model, instead, the direction for which neurons code is independent of the agent's heading direction (where it is moving to) or bodily coordinate (how it is posed or which direction it is facing). Rather, the neurons code for a direction in a world-centric polar coordinate. Thus, the centre of the receptive field of neuron $i$ is an angle $\alpha_i = \frac{i-1}{N_s}2\pi$ with respect to an absolute reference frame independent of animal's orientation, e.g., a world-centric east (positive $x$-axis). This can be achieved, for instance, by anchoring to one or more external cues, such that, as the agent moves in space, the ring-attractor network does not rotate with the agents' body axis (see Fig. 1C). Clearly, this does not mean that all the agents necessarily share common allocentric reference frames; how the agents define the zero of their coordinate (and thus how they define, e.g., north) is a matter of indexing the neurons and is inconsequential for their neural dynamics and its resulting movement pattern. Thus, anchoring to different external cues (or not having a consensus on which direction is north) does not affect the collective movement of the agents.

To further clarify how an allocentric representation of space can be achieved, we can write, $\hat{\alpha}^{\text{allo}} = \hat{\alpha}^{\text{ego}} + \mathbf{H}^{\text{allo}}$, where $\mathbf{H}^{\text{allo}}$ is the agent's heading direction in an external (allocentric) reference frame (an external polar reference frame not to be confused with the agent's

allocentric reference frame). It is known that animals can maintain such an allocentric representation of their heading or head direction using ring-attractor networks (for instance, via path integration combined with the utilization of environmental cues), which they utilize to maintain an allocentric representation of space[48,55-58,70]. To do so, in our allocentric model, we have assumed the agent has such a capacity and encodes polar directions in an allocentric way.

We take the synaptic connectivity of the network, $J_{ij}$, to be a modified cosine function, as follows:

$$J_{ij} = \cos\left(\pi\left(|\alpha_i - \alpha_j|/\pi\right)^{\nu}\right) \tag{2}$$

Here, $\alpha_i$ and $\alpha_j$ refer to the position of neural groups on the ring. Eq. (2) implies that neurons in the network have periodic connectivity and endow the network with a ring structure. With $\nu = 1$, positive and negative synapses are found in roughly equal numbers, and for $\nu < 1$, the network connectivity is locally more excitatory and globally more inhibitory, which requires more inhibitory synapses to exist in the system.

We assume the network dynamic is governed by a Hamiltonian, as follows:

$$H = -\left[\frac{1}{N_s}\sum_{i,j}J_{ij}\sigma_i\sigma_j + \sum_i(h_i\sigma_i - h_b\sigma_i)\right]. \tag{3}$$

Here, $h_b$ is a constant term that promotes inhibition of the network activity. Assuming neurons favor a state with the lowest energy, this Hamiltonian implies that each neural group tends to assume a state favored by its input. We use the Glauber dynamics to simulate the network's dynamics[94]. At each step, a neuron is chosen at random, and the energy difference resulting from updating the neuron's state is calculated. The neuron's state is flipped with certainty if the energy difference becomes negative, and it is flipped with probability $\exp(-\beta\Delta H)$ if the energy difference is positive. We repeat the Glauber dynamics for $T_0 N_s$ steps for the network to equilibrate. After this, we update the agents' position according to the equilibrium activities of the neurons. In this stage, the agent moves with a speed vector $\mathbf{v}$ determined by the activity of its neural network according to the following equation:

$$\mathbf{v} = v_0/N_s \sum_{i \in \text{ active spins}} \hat{\alpha}_i. \tag{4}$$

Where $\hat{\alpha}_i$ is the egocentric or allocentric goal vector pointing toward direction $\alpha_i$ (and is the same as the center of the receptive field of the neural group).

The extension of the model of individual movement and information acquisition to a model of collective movement is rather straightforward and only requires a change of perspective: it is enough to allow several such agents to perceive each other as possible targets and interact. We consider three variants of such a model of collective motion, based on the regulation of social interaction.

In the baseline model, we consider the simplest case, where each agent is a target to other agents, with a magnitude of external field, $h_0^s$. We usually report total social attraction, defined as $h_t^s = Nh_0^s$. We also consider two variants of this baseline model. In the model with short-range repulsion, the amplitude of the receptive field is a step function of the distance of the focal agent to the target. Below a collision radius, the amplitude of the external field is taken to be a negative value, ensuring conspecifics act as a repelling stimulus, rather than an attracting one. We note that this is consistent with the recent finding that flies maintain a neural representation of their anti-goal (goal direction $+180°$)[41]. Taking the magnitude of social repulsion large enough ensures agents avoid a collision. Above the collision radius, the external field is taken to be positive, ensuring conspecifics are attracting stimuli.

In the second variant of the model, we study the distance dependence of social attraction. In this variant, the amplitude of the receptive field is taken to decay with the distance between the focal agent and its target, $d$, according to an exponential, $h_0^s \exp(-d/\zeta L)$, where $L$ is the linear size of the space. With this choice, for $d < \zeta L$, the exponential term is approximately a constant and equal to 1. $\zeta L$ is thus the characteristic length of social attraction, above which the strength of social attraction decays exponentially fast.

**The neural field model.** Our neural field model is based on Amari's classical formulation[86]. We consider a neural network with a ring structure in which neurons are arranged on a ring with a modified cosine-shaped synaptic connectivity, given by Eq. (2). The dynamics of the network are governed by the following adaptation of Amari's neural field dynamics to a discretized one-dimensional network:

$$\frac{d u_i(t)}{dt} = - u_i(t) + \frac{1}{N_s} \sum_{j=1}^{N_s} J_{ij} F\left( u_j(t) \right) - h_b + h_i(t), \quad (5)$$

Here, $u_i(t)$ denotes the membrane potential of neuron $i$ at time $t$. $h_b$ is the global inhibition on the network, and $h_i(t)$ is the external current on neuron $i$, originating from sensory input. We have taken the output function of neurons to be a hyperbolic tangent function parameterized by a slope parameter $\beta$, $F( u_i(t)) = \tanh(\beta u_i(t))$. $\beta$ can be thought of as an inverse noise parameter: For large $\beta$, the output becomes steeper and more sensitive to input, while for small $\beta$, it represents noisier dynamics where small differences in the input do not build up large output differences.

We note that the intensity of connections between neuron $i$ and $j$ (synaptic connectivity) can in general depend on time lag, $J_{ij}(t - t')$. Such a time-dependence can allow us to take pulse conduction or synaptic delay into account[86]. While incorporating such a memory-based representation of neural dynamics may provide additional insights, here we do not investigate the consequences of synaptic delay.

The sensory input on neuron $i$ originates from a Gaussian, $h_i = h_0 \exp[-\frac{|\hat{\alpha}_i - \theta_{target}|^2}{2\sigma^2}]$. We consider both egocentric and allocentric representations of space as explained before. With an allocentric representation, the direction vector, $\hat{\alpha}_i$, refers to an allocentric direction in a reference frame independent of the agent's heading direction or orientation. On the other hand, with an egocentric reference frame, the reference frame rotates with the individual's heading direction.

Finally, each neuron encodes for movement along the same direction that it receives sensory input from, such that the agent's speed (goal vector) is determined based on positive neuron activities, according to:

$$\mathbf{v} = \frac{v_0}{N_s} \sum_i \max\left(0, \tanh(\beta u_i)\right) \hat{\boldsymbol{\alpha}}_i. \quad (6)$$

Thus, both the direction of the movement and the magnitude of the agent's speed are shaped by the collective activity of the neurons.

The extension of the model to a model of collective motion is straightforward and is done as before, by making each agent a target for each others' ring-attractor network.

Switching between reference frames. In the extension of our model to consider switching between reference frames, we assume the agent can continuously switch between the two reference frames, such that it does not temporarily lose its sense of orientation when switching. We term this switching scenario continuous switching between reference frames. Explicitly, we assume that when the agent switches from the allocentric to the egocentric reference frame, its current heading becomes the zero of its egocentric reference frame:

$$\text{Switch from allo to ego at time } t : \quad O^{ego,(a)}(t) = H^{(a)}(t - 1) \quad (7)$$

Here, $H^{(a)}$ refers to the agent a's heading direction in an external, allocentric reference frame (which, in terms of its origin (zero), can be different from the agent's allocentric reference frame), and $O^{(a)}$ refers to the zero of the agent's reference frame. This is a realistic assumption following naturally from the very definition of an egocentric frame, according to which, the frame can only depend on the agent's body axis. Similarly, when the agent switches from an egocentric to an allocentric reference frame, its current heading is defined to be the (new) origin (zero) of its allocentric reference frame:

$$\text{Switch from ego to allo at time } t : \quad O^{allo,(a)}(t) = H^{(a)}(t - 1) \quad (8)$$

This assumption ensures that the same bump of activity along the ring does not lead to a motion along different directions before and after a switch. Intuitively, this means that the agent does not lose its sense of orientation while switching and can effectively relate egocentric and allocentric perceptions of space, without getting lost. Biologically, such an ability to change the allocentric reference frame, which we refer to as re-anchoring, can result from maintaining a sense of orientation via exploiting external cues[48], path integration[48,50], or memory-based cues[46].

A minimally modified model based on only direction coding. In the Supplementary Information (S.13), we also present results for a model in which the agent always moves with a constant speed, but its goal direction, and thus, heading direction, is determined by the ring-attractor network, and show that this model gives rise to similar results regarding the emergence of collective motion.

In this modified model, at each time step $t$, the agent's heading $H(t)$ is computed from the positive neuron activities. Defining:

$$C_x(t) = \sum_{i=1}^{N_s} \max\left(0, \tanh(\beta u_i(t))\right) \cos(\alpha_i), \quad (9)$$

$$C_y(t) = \sum_{i=1}^{N_s} \max\left(0, \tanh(\beta u_i(t))\right) \sin(\alpha_i), \quad (10)$$

the agent, $a$'s heading is:

$$H^a(t) = \text{atan2}\left(C_y, C_x\right), \quad (11)$$

wrapped to $[0, 2\pi]$. Here, as before, in the allocentric case, $\alpha_i$ is defined in a world-centric reference frame (so the ring-attractor network, being anchored to external cues, does not rotate as the agent moves in space). In the egocentric case, $\alpha_i$ is defined with respect to the agent's current heading so that the network rotates with the agent's heading direction.

## Statistics and reproducibility
**Simulations.** The base parameter values used for the individual motion patterns are as follows: $N_s = 100$, $v_0 = 10$, $\sigma = 2\pi/N_s$ (unless otherwise specified). All the simulations are performed in a space with periodic boundaries. Unless otherwise stated, the linear size of the space is equal to $L = 1000$. For collective movement, agents see each other, such that each agent is a target to every other agent's ring-attractor network with an amplitude of external field equal to $h_0^s$. We report the total external field defined as $h_t^s = h_0^s N$, where $N$ is the population size. In Fig. 3G, H, and I $N_s = 400$ and other parameter values remain the same. The averages and error bars in Fig. 3H and I are calculated based on the stationary state of a sample of 5 runs for 10,000 timesteps. The target's speed along the x- and y- axis obeys a random walk with speed $v_t$, shown on the panels. In Fig. 3G, we have used 80 simulations, and the simulations stop when the agent reaches a close proximity of the target (5 units). Here, the target is stationary. The use of a larger sample is due to the fact that in such a speed decision-making task, it is not possible to rely on long-time stationary

trajectories to provide stronger statistics. The amplitude of the external field in all the cases is equal to $h_0 = 0.0025$. Error bars represent the standard deviation over the sample. Large error bars for large values of $\beta$ in Fig. 3G are because the agent's decision-making accuracy decreases for too large values of $\beta$, while its speed increases. Thus, while in some trials the agent reaches the target rapidly by moving directly toward the target, in other trials it starts by moving in the wrong direction.

In Fig. 4, a sample of three simulations run for 10,000 timesteps in a population of $N = 80$ agents moving in a space with periodic boundaries and linear size $L = 1000$ is used. In Fig. 5, simulations are performed for 15,000 timesteps. A sample of 10 simulations is used to calculate the distribution. The distributions are calculated based on the last 10,000 timesteps of the simulations to ensure stationarity. Here, $N = 80$ and $L = 1000$.

In Figs. 6 and 7, simulations are performed for 15,000 timesteps. A sample of 5 simulations is used to calculate the distribution. The distributions are calculated based on the last 13,500 timesteps of the simulations to ensure stationarity. Here, $L = 1000$, and $N = 10$ and $N = 320$ respectively.

The base parameter values used in the neural field model (Figs. 9 and 10) are: $N_s = 100$, $v = 0.5$, $v_0 = 0.05$, $\sigma = 0.4$, $h_b = 0$, $N = 80$, $\Delta t = 0.3$, $\beta = 1000$, and $L = 1000$. To simulate these dynamics, we implement Euler integration, by discretizing time according to:

$$u_i^{(a)}(t + \Delta t) = u_i^{(a)}(t) + \Delta t \left[ -u_i^{(a)}(t) + \frac{1}{N_s} \sum_{j=1}^{N_s} J_{ij} \tanh\left(\beta u_j^{(a)}(t)\right) - h_b + h_i^{(a)}(t) \right]$$

(12)

Here, $a$ refers to agent $a$. Using this discretization, at each simulation step, the agents' neural networks are updated, and they move synchronously. In Fig. 9, the results show a time average and ensemble average over a sample of 5 simulations run for 30,000 timesteps timesteps (time-averaged over the last 5000 timesteps). In Fig. 10A and B, a sample of 5 simulations run for 20,000 timesteps is used. A time average over the last 5000 timesteps is taken.

**Measures of collective movement.** In this section, we define the measures used to analyze simulation data in our study. Further analysis is performed in the Supplementary Information and a more complete list of the measures used in the study is provided in the Supplementary Information (S.15).

• Global Order: As a measure of global order, we have used the angular order parameter. This is calculated as the sum of the normalized velocity vectors of the individuals. To do so, we have normalized direction vectors to 1 and then summed over all individuals' normalized vectors. Namely,

$$\text{GO}(t) = \left\| \frac{1}{N} \sum_{i=1}^{N} \hat{\mathbf{v}}_i(t) \right\| = \sqrt{\left(\frac{1}{N} \sum_i \frac{v_{x,i}(t)}{\|\mathbf{v}_i(t)\|}\right)^2 + \left(\frac{1}{N} \sum_i \frac{v_{y,i}(t)}{\|\mathbf{v}_i(t)\|}\right)^2}.$$

(13)

Here, $\mathbf{v}_i(t)$ is the velocity vector of individual $i$ at time $t$, and $\|\mathbf{v}_i(t)\| = \sqrt{v_{x,i}(t)^2 + v_{y,i}(t)^2}$. The unit velocity vector is defined as

$$\hat{\mathbf{v}}_i(t) = \begin{cases} \frac{\mathbf{v}_i(t)}{\|\mathbf{v}_i(t)\|}, & \|\mathbf{v}_i(t)\| > 0, \\ \mathbf{0}, & \|\mathbf{v}_i(t)\| = 0. \end{cases}$$

(14)

Values of GO close to 1 indicate strong alignment, and values close to 0 indicate weak alignment. We note that, in practice, the minimum of this quantity approaches zero only in the limit of infinite population size.

• Local Order: As a measure of local order, we have used the topological vectorial order parameter. This is a measure of the average direction of the velocity vectors of individuals within a local neighborhood defined based on topological distance. Namely, let $\mathcal{N}_i^{(k)}(t)$ be the set of the $k + 1$ nearest agents (including $i$) to agent $i$. Then

$$\text{LO}(t) = \frac{1}{N} \sum_{i=1}^{N} \left\| \sum_{j \in \mathcal{N}_i^{(k)}(t)} \hat{\mathbf{v}}_j(t) \right\|.$$

(15)

This quantity takes a value up to $k + 1$. To achieve a normalized local order parameter we have divided this value by $(1 + k)$ so that the maximum local order is 1. Besides, the minimum of this value is always larger than 0 and approaches zero as $k$ increases (because the sum of $k$ random vectors becomes small only in the limit of $k \to \infty$). We have set $k = 5$. The results are valid for other reasonable choices. A high value indicates strong alignment (coordinated movement in a common direction), while a low value indicates weak alignment.

• Mean distance between all pairs: This is the average distance between all the pairs in the population, $\sum_{i,j} d_{i,j}/(N(N-1))$, where $d_{i,j}$ is the distance between individuals $i$ and $j$, and $N$ is the number of agents in the population.

## Supplementary videos

Supplementary videos (SV) 1 to 8 show examples of collective motion patterns using the spin system model. Parameter values used in these videos are as follows: $N_s = 100$, $v_0 = 10$, $\sigma = 2\pi/N_s$, $h_b = 0$, $\beta = 400$, and $L = 1000$.

Supplementary videos SV.1 to SV.6 present the dynamics of collective motion over time. Snapshots of the Videos are presented in Figs. 6 and 7. In SV. 1 to SV. 3 $N = 10$ agents interact, and in SV. 4 to SV. 6 $N = 320$ agents are considered. Total social attraction in these videos is equal to $h_t^s = 0.024$, $h_t^s = 0.16$, and $h_t^s = 0.28$, respectively. These videos correspond to snapshots presented in Fig. 6E–G, respectively. In Both SV.1 and SV.2 a variety of motion patterns including intermittent swirling, sudden direction change, and fission-fusion dynamics can be observed. SV.3 is chosen close to the phase transition between the collective motion-aggregation phase, and the intermittency between these two modes of motion can be observed.

SV.4 to SV.8 show examples of collective motion in large groups. The total social attraction is set equal to $h_t^s = 0.02$, $h_t^s = 0.032$, $h_t^s = 0.08$, $h_t^s = 0.12$, and $h_t^s = 0.16$, respectively. SV.5, SV.7, and SV.8 correspond to snapshots in Fig. 7E–G, respectively. Both SV.4 and SV.5 show strong collective motion. However, intermittency and coexistence of different modes of motion, such as fission-fusion dynamics, swirling, startling, and sudden direction changes, can be observed. Similarly, in SV.6 and SV.7 a variety of collective motions can be observed. Explosive and implosive motion of the group leads to highly coordinated state changes between different motion patterns. The explosive and implosive motion is stronger in SV. 8 chosen at the transition between collective motion-aggregation.

Supplementary videos SV.9 to SV.16 show examples of collective motion in the neural field model. SV.9 to SV.12 show the collective movement patterns in a community of $N = 80$ agents with an allocentric representation of space. In SV.9 to SV.12, total social attraction is set to, $h_t^s = 0.08$, $h_t^s = 0.24$, $h_t^s = 0.28$, $h_t^s = 0.32$, respectively. Other parameter values are: $N_s = 100$, $v_0 = 0.05$, $\sigma = 0.4$, $h_b = 0$, $\beta = 1000$, $L = 1000$, and $\Delta t = 0.3$.

SV.9 shows an example of collective motion patterns in the neural field model in the collective motion phase for small $h_t^s$, characterized by unstable dynamics and intermittency between different movement patterns, such as directed motion, flash expansion, sudden direction change, and fission-fusion dynamics. SV.10 and SV.11 show examples of collective motion patterns in the collective motion phase for larger values of $h_t^s$. The population shows transient motion between a variety

of patterns, such as milling, moving bands, and fission-fusion dynamics. SV.12 shows collective motion patterns close to the collective motion-aggregation phase transition. The system spends more time in a state where the population is composed of subgroups of coherently moving individuals, with high alignment, exhibiting directed motion.

In SV.13 and SV.14 we show collective motion patterns in larger groups of $N = 320$ individuals. Here, $h_t^s = 0.24$, $h_t^s = 0.28$, respectively. In SV.13, intermittency between a variety of collective motion patterns, from milling to fission-fusion, flash expansion, sudden direction change, and highly ordered motion is observed. SV.14 is chosen closer to the collective motion-aggregation phase transition. Hereafter, an initial period of mostly milling, the population forms subgroups of mobile aggregates moving with high order.

SV.15 shows an example of collective motion when individuals randomly employ an allocentric and an egocentric representation of space. Here $h_t^s = 0.08$, corresponding to SV.9, where the same parameter values, but for a purely allocentric representation of space are used. Here, at each timestep, individuals randomly employ an allocentric or egocentric representation of space. The probability of being in the egocentric state is $\omega = 0.8$, corresponding to the maximal global order region for small $h_t^s$ in Fig. 10. An example snapshot of this regime is shown in Fig. 10D. Switching between the two states stabilizes highly ordered collective motion. Although such movement patterns are also observed when individuals employ a purely allocentric representation of space, these patterns are not stable in the absence of a switch to egocentric. This, in turn, leads to the esthetic appeal and complexity of collective motion patterns in purely allocentric flocks, due to intermittency between different movement patterns.

SV.16 shows collective motion patterns for $h_t^s = 0.4$ and $\omega = 0.2$. With a purely allocentric representation of space, in this regime, often milling is observed (for slightly smaller $h_t^s$, as in SV.12, transitions between milling and moving aggregates are observed). Such a milling pattern can be seen in the initial times in the video. However, a small probability of being in the egocentric state destabilizes milling and leads to highly ordered motion of moving aggregates.

SV.17 shows the dynamics of the model in the absence of recurrent connections and with an allocentric representation of space. Here, the same parameter values as in SV.9 to SV.16 are used. However, $\beta = 100$ and $h_t^s = 0.36$, and $L = 100$. In this Video, we have decreased the arena size due to the fact that in the absence of recurrent connections, the agents move too slowly. To remove the recurrent connections, using the neural field model, we have set $J_{ij} = 0$ for all $i$ and $j$. This ensures all other aspects of the model are preserved. However, the dynamics of the system are the result of feedforward connections. As can be seen in the Video, starting from random initial positions, agents move toward each other in an accelerating fashion and coalesce in the same position with slow, random walk-like movement. This shows that recurrent connections are essential for the rich dynamical patterns observed in the model.

## Data availability

The raw MATLAB data generated in this study have been deposited in Figshare under accession code https://doi.org/10.6084/m9.figshare.28925888[111]. The processed data generated in this study are provided in the Manuscript and its Supplementary Information.

## Code availability

MATLAB codes used in the manuscript can be accessed via Code Ocean and the Supplementary Data 1.

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

## Acknowledgements

We acknowledge Yasser Roudi for insightful discussions and Maya Dagher for proofreading an earlier version of the manuscript. The authors acknowledge funding from Deutsche Forschungsgemeinschaft (DFG, German Research Foundation) under Germany's Excellence Strategy—EXC 2117-422037984 (I.D.C.), the Deutsche Forschungsgemeinschaft Gottfried Wilhelm Leibniz Prize 2022 584/22 (I.D.C.), the European Union's Horizon 2020 Research and Innovation Program under the Marie Skłodowska-Curie Grant agreement no. 860949 (I.D.C.), the Pathfinder European Innovation Council Work Program no. 101098722 (I.D.C.), the Office of Naval Research Grant N0001419-1-2556 (I.D.C.).

## Author contributions

M.S. and I.D.C. designed the research, M.S. performed the research, and M.S. and I.D.C. wrote the paper.

## Funding

## Competing interests

The authors declare no competing interests.
