## [Transparent Peer Review file · Nature Communications]

Allocentric Flocking

Corresponding Author: Dr Mohammad Salahshour

Version 0:

Reviewer comments:

Reviewer #1

(Remarks to the Author)

Review for "Allocentric Flocking" by Salahshour and Couzin

In this paper, the authors proposed a model for studying the group-level spatial dynamics arise from the interactions of individuals in the group. The key new idea was to assume that individual animals encode their headings in an allocentric coordinate system. The authors found that, under certain assumptions (including that individuals themselves act as sensory inputs to each other), coherent collective movement patterns could emerge in this model. The model enables effective tracking of dynamically moving target. They also studied an alternative model in which the heading was encoded in an egocentric coordinate system, and they found that effective tracking of dynamically moving target was absent.

The paper contains an interesting idea. The idea of grounding the key navigation variables using knowledge of neural circuits when studying collective behavior is conceptually appealing. Although the modeling framework is based on recent studies by some of the same authors (thus not entirely new), the study of the implications of different reference frames on the implied collective behavior appears to be novel. The comparisons between different versions of models are thorough. Additional analysis in the supplementary materials further consolidate the results in the main text.

I do have several substantial concerns. First, I find the current version of the paper to be difficult to read. Also, several aspects of the modeling can be improved. A number of important questions, some of which are conceptual, need to be clarify. At the moment, although I think the manuscript reports several interesting results, it is somewhat difficult to judge the significance of these results. I hope these issues can be addressed in a revision to make it easier to judge the significance and relevance of this paper.

*****Substantial concerns:

1. Clarification of the definition of heading direction

Can the authors clarify what they meant by heading (or heading direction)? I initially thought it meant the head direction, i.e., the direction of the head is facing. The reason why I made that interpretation was because that the ring attractor models used prominently in the manuscript had been studied extensively in the context of head direction cells (not "heading direction cells"). Further reading of the manuscript suggest that the authors may be referring to the direction of the movement (heading).

The head direction (the direction the animal is facing) and the movement direction (the direction the animal is moving) is generally different.

If it was the former, head direction would be a much better terminology. If it was the latter, heading direction would be the accuracy terminology. However, in this case, the reference to the ring attractor model would seem to be misleading and need to be further clarified. The reason is that ring attractor models and many of the neuroscience references cited in the manuscript were studying the encoding of head direction, but not the direction the animal is heading toward.

2. The formulation of the model needs improvements

I found the formulation of the model to be a bit problematic and not entirely satisfying. Currently, the model is specified by the Hamiltonian, and Glauber dynamic is used to simulate the network dynamics. The influence of the external sensory cues was attached to the network in a rather post-hoc way.

Thus, this model is not directly constructed based on a group of neurons and the connectivity. I am concerned that the simulated dynamics may not faithfully reflect the network dynamics of a more realistic network model.

The paper would be strengthened if a fully-consistent and realistic neural network model could be constructed and simulated, by coupling a set of ring attractor network modules studied in neuroscience, such as the classic model by Zhang (1996). Note that in the Zhang model (as well as many other versions of the ring attractor models), one can explicitly write down the dynamical equations governing the dynamics of individual neurons in terms of membrane potential and firing rate. Neurons encoding sensory inputs can be wired together with the neurons on the ring by defining specific connectivity or through some learning rules (e.g., Hebbian plasticity).

Zhang, Kechen. "Representation of spatial orientation by the intrinsic dynamics of the head-direction cell ensemble: a theory." *Journal of Neuroscience* 16.6 (1996): 2112-2126.

3. Issues with allocentric v.s. egocentric representations

A central aspect of the work is the comparison between models based on the allocentric v.s. egocentric representation. There are several concerns related to this point that I hope the authors could address.

- First, how the egocentric-based and allocentric-based representations were modeled needs a better explanation. The descriptions in the Method section on page 11 and 12 are somewhat ambiguous. The main difficulty is the model is described by a hybrid of quantitative and qualitative components. It would be helpful if the authors could define the full model by writing down the equations for the allocentric version of the model, then do the same thing for the egocentric version of the model.

- Second, there is substantial evidence suggesting that the brain maintains multiple reference frames. It may well be the case that the brain generate representations of the environment both in allocentric sense and in egocentric sense simultaneously. In the authors' framework, it felt that one was forced to choose one or another. How can the results presented in the paper reconcile with the coexistence of multiple reference frames?

- Third, the authors seemed to consider the allocentric reference frame as being fixed. What if the neural system re-anchors to a different sensory cue? (Re-anchoring is not uncommon.)

- Fourth, the results based on the comparison of the egocentric-model and allocentric-model paint a pretty complex picture. My understanding of the results is that allocentric representation only convey an advantage in certain regimes. For example, in section "Finding and staying close to a static target", the egocentric representation in fact can have an advantage. The Abstract should be revised to more accurately reflect these results.

4. The assumption on the relationship between the movement and neural activity needs justification.

One crucial assumption of the models proposed was that "the agent's movement is dictated by the activity of its neural network, where the velocity vector is computed based on the activity of the neurons." It is not obvious to me why this is a good assumption.

In previous ring attractor networks, the activity bump on the ring typically encodes the current stimulus, rather than the decision or movement. In particular, this is the case for the head direction system. Any experimental evidence that the authors could provide? In any case, it would be helpful if this important assumption can be justified or discussed in more detail.

5. The functional importance of attractor dynamics needs to be further clarified.

The authors motivated the ring-attractor model based on the literature of the ring-attractors in the fly and rodent head direction system. This makes sense. But is the continuous dynamics actually important for the results in the paper?

The paper would be strengthened if the authors can construct alternative models that are similar to these studied in the paper, but without attractor dynamics. Right now, the role of attractor dynamics is unclear.

It is possible that attractor dynamics is not essential. If that's indeed the case, it would lead to a different interpretation of the results, and likely also a simplified account of these results (perhaps a model without a neural circuit architecture would be sufficient to explain the results)? Looking into these issues would help distill the key factors in the model that leads to the key predictions.

6. Experimental relevance of these results remains unclear

The connection to experimental results is only briefly mentioned in the paper. To understand the implication and significance of these modeling results, a more direct and systematic comparison to the experimental literature would help a lot. I am not sure if quantitative comparison is possible in the case. But a more careful treatment of when the models generate consistent or inconsistent predictions with the data would benefit the paper.

Other concerns/suggestions for improvements:

** In the Introduction, it is stated that "it has recently been shown that ring attractor networks also account for how fruit flies, locusts and zebrafish (and thus both invertebrates and a vertebrate) make decisions when choosing between multiple spatial "targets"..." The references cited there were modeling studies. Is there any neuroscience evidence suggesting that the ring attractors actually encode the decision of head turning? My understanding of the literature is that head direction cells in the fly and rodent are considered to encode the current head direction, not the turning decision. But I could be missing something. (This is related to Point # 4 above)

** I felt the flow in the Results section could be improved. It was a bit slow to get to the key results. Perhaps combining and compressing the subsections "Spontaneous neural dynamics" and "From spontaneous neural dynamics to individual motion" would help.

**It would be useful to label/number the subsections in the Results section.

** How does the connectivity pattern affect the results? For example, if varying the width of the connectivity profile, would that affect the results?

** In the Method section, it would help if the equations for defining VOP and geometric VOP could be included.

Suggestions for the Figures:

I also have a list of suggestions regarding the presentation of the figures/figure legends.

Fig. 1B: I think the cartoons for the two insects meant two different individuals. If that is the case, it would be useful to label them. Otherwise, some may also interpret it as an insect moves to a different spot with the head direction fixed.

Fig. 2: in the second and third columns, please plot and label the axes. What spaces were these plots based on? 2-D physical space? This may not be as obvious. Also, please clarify the meaning of "200 units". What is the unit?

Fig. 3G,H,I: please define and explain the quantities (on the y-axis) being plotted in the figure legend/caption.

Fig. 4: It would be useful to add (sub-)titles in individual figure panels to indicate the allocentric v.s. egocentric cases.

Fig. 5: This figure is difficult to interpret on its own without going back and forth together with the text. Please improve the figure legend and explain what was plotted. In particular, it would be useful to remind the readers the meaning of the variables being plotted.

Fig. 8: In the legend, it is stated that "B and C: The motion pattern (A) and the neural activity (B) of a subgroups of 3

coherently moving agents among the 80 agents presented in A are shown." This is confusing. How is Fig 8B showing the neural activity? I thought that should be panel C and E. I think there might be an error here.

Fig. 9: would it be possible for the authors to add another set of curves for $L=100$?

Having results based on parameters that spans several orders of magnitude would make the argument more convincing.

(Remarks on code availability)

Reviewer #2

(Remarks to the Author)
see attachment

(Remarks on code availability)

Version 2:

Reviewer comments:

Reviewer #1

(Remarks to the Author)
Review for the resubmission of "Allocentric Flocking"

I would like to thank the authors for their detailed response to my original review. To address the critiques raised in the first round of review, the authors made substantial changes to the paper.

The authors added a new model, i.e., neural field model, and analyzed and compared the prediction of this model with the original model based on a spin system. Furthermore, the authors extended their analysis to allow the switching between the allocentric and egocentric reference frameworks. The authors also make a few other improvements.

These additional analyses substantially improved the paper, and addressed the major concerns in my initial review. I only have a few relatively minor concerns, mostly about the presentation of the paper.

First, while the revised version of the Introduction made the modeling framework more grounded on the neuroscience literature, now it is very long now and the logic is a bit convoluted. I think the paper would benefit by streamlining the Introduction.

Second, are there ways to empirically adjudicate between the two versions of the models (spin system and neural fields)? I feel the paper would benefit from a discussion on this issue. In addition, the similarities and differences in the predictions of the two models could be made more clear to better orient the readers.

Third, the authors show that the models exhibit different collective patterns under different parameter regimes. This work mainly comes from mechanistic considerations. Yet what might be the functional benefits of these different patterns? Do these considerations help us understand why certain patterns are observed in certain scenarios for certain species? I think readers would be interested in hearing if there is any insight the authors may have. So a brief discussion on the functional aspects of the results might be useful.

(Remarks on code availability)

Version 3:

Reviewer comments:

Reviewer #1

(Remarks to the Author)
The authors have done a nice job addressing my concerns, and the revised paper is further improved.

This paper represents a step toward better understanding how the navigation circuits may support collective motion. I

recommend the publication of this paper.

(Remarks on code availability)

Responses to review comments

1 Overview of the changes

We thank both referees for their constructive feedback which we have used to significantly improve our manuscript. Below is an overview of the changes made in the revision

1. Additional (neural field) model. In addition to our original “spin system” formulation, we have introduced a new “neural field” model based on Amari’s classic framework. This one-layer attractor network faithfully reproduces—and in some respects enriches—the phenomenology of allocentric versus egocentric navigation, while embedding our approach more firmly in the neuroscience literature.

2. New Analysis of Frame Switching. Responding to Reviewer 1’s suggestion, we have explored the effects of probabilistic, rapid switching between allocentric and egocentric references, demonstrating how intermittent re-anchoring can further stabilize global order while attenuating spatiotemporal complexity.

3. Expanded Conceptual Framework. We now provide a more clear and biologically grounded discussion clarifying the aim and scope of our work, clarifying how allocentric and egocentric representations can be implemented, how they may switch or coexist, and how those choices shape emergent collective dynamics.

4. Empirically Grounded Discussion. We have strengthened the manuscript’s connections to experimental work across fish, birds, insects, and mammals, explicitly comparing allocentric flocking predictions to empirical findings.

5. Enhanced Presentation and Terminology. We have clarified neuroscience terminology, reorganized and numbered Results subsections, improved figure legends, and captions, and added new citations to better align our work with both the cognitive-neuroscience and collective-behavior communities.

Below, we provide our point-by-point responses to both reviewer’s comments.

2 Response to Reviewer N. 1

Reviewer N. 1:

In this paper, the authors proposed a model for studying the group-level spatial dynamics arise from the interactions of individuals in the group. The key new idea was to assume that individual animals encode their headings in an allocentric coordinate system. The authors found that, under certain assumptions (including that individuals themselves act as sensory inputs to each other), coherent collective movement patterns could emerge in this model. The model enables effective tracking of dynamically moving target. They also studied an alternative model in which the heading was encoded in an egocentric coordinate system, and they found that effective tracking of dynamically moving target was absent.

The paper contains an interesting idea. The idea of grounding the key navigation variables using knowledge of neural circuits when studying collective behavior is conceptually appealing. Although the modeling framework is based on recent studies by some of the same authors (thus not entirely new), the study of the implications of different reference frames on the implied collective behavior

appears to be novel. The comparisons between different versions of models are thorough. Additional analysis in the supplementary materials further consolidate the results in the main text.

I do have several substantial concerns. First, I find the current version of the paper to be difficult to read. Also, several aspects of the modeling can be improved. A number of important questions, some of which are conceptual, need to be clarified. At the moment, although I think the manuscript reports several interesting results, it is somewhat difficult to judge the significance of these results. I hope these issues can be addressed in a revision to make it easier to judge the significance and relevance of this paper.

Response:

We thank you for this constructive feedback. We apologize for the lack of clarity. We have significantly revised our manuscript to address your comments and to improve the manuscript. We have also made the novel aspects of our approach and findings more clear, as detailed below.

Reviewer N. 1:

1. Clarification of the definition of heading direction

Can the authors clarify what they meant by heading (or heading direction)? I initially thought it meant the head direction, i.e., the direction of the head is facing. The reason why I made that interpretation was because that the ring attractor models used prominently in the manuscript had been studied extensively in the context of head direction cells (not “heading direction cells”). Further reading of the manuscript suggest that the authors may be referring to the direction of the movement (heading).

The head direction (the direction the animal is facing) and the movement direction (the direction the animal is moving) is generally different.

If it was the former, head direction would be a much better terminology. If it was the latter, heading direction would be the accuracy terminology. However, in this case, the reference to the ring attractor model would seem to be misleading and need to be further clarified. The reason is that ring attractor models and many of the neuroscience references cited in the manuscript were studying the encoding of head direction, but not the direction the animal is heading toward.

Response:

Thank you for making this important point, which points towards some of the novel aspects of our study (which required better clarification).

To address this comment, in the revised manuscript we have differentiated three concepts, head direction, heading direction, and goal direction. As the referee notes, ring attractor networks have been predominantly used to address head direction: how animals can maintain an allocentric representation of their heading or head. Our study has a different goal and scope.

Although we borrow the ring-attractor formalism familiar from head-direction (HD) studies, our network does not represent head direction. Indeed, as we detail below, in the context of allocentric representation of bearings to objects, the animal need not have any representation of body, head, or any other axis. Furthermore, Our work focuses on how animals integrate sensory information to establish their goal direction. Without loss of generality we assume that animals have the capacity to turn towards their goal direction. Therefore, we use the ring attractor as a mathematical device (albeit one that is grounded in its biological reality in navigational circuits) to link sensory input to the establishment of a goal direction.

In the revised Introduction, we clarify head direction and heading direction as follows (lines 46 to 55):

Multiple interconnected and intercommunicating ring attractor networks coexist in the brain. Central to spatial navigation is animals’ neural “compass”, often termed their “heading compass” or “head

compass”, in which the cellular activity rotates as the animal changes heading, allowing estimation of body/head orientation relative to visual [60] (and in some species also magnetic [61–63]) cues. Prominent visual cues employed to “tether” the compass include polarized light [64–66], the sun [60, 61] and prominent distant, and therefore relatively stationary, objects in the environment [61, 62]. In this way, the animal can maintain an allocentric reference frame for its heading, i.e., its orientation with respect to external cues [43, 48, 55, 67–72]. While some species, such as fish [56], have rigid bodies, in others, such as mammals, animals can maintain head direction in addition to heading [57, 58]. We note that head and heading directions have yet to be disambiguated in insects [54, 55].

Then, we introduce the concept of “goal direction” (lines 71 to 77):

*In addition to their heading compass, animals have also been found to encode their “goal direction” in a ring attractor network. Here, the bump of activity represents the desired direction of travel for the animal at the present moment in time [41, 43, 44, 53]. While it is known that animals turn towards their goal vector during navigation, with the neural circuitry responsible for converting allocentric goals into appropriate egocentric steering controls having been dissected in *Drosophila* [41, 53], relatively little work has been yet conducted into how the goal vector is itself established when there are multiple alternatives [77].*

And then, clarify the aim of our study in using ring attractor network (lines 78 to 82):

Here, we focus our attention on this less-explored aspect of decision-making and make the reasonable assumption that animals can turn towards their goals. Thus we do not explicitly model how animals maintain their allocentric heading, which they are known to be able to do, but rather how sensory information—with a specific focus on visual information—may be integrated to create time-varying goal directions.

Thus, we do not aim to address how animals maintain an allocentric representation of space or how they steer towards their goal, which is (empirically) known that they can do. Rather, by making the reasonable assumption that animals have these abilities, we address how animals establish a goal direction by transforming sensory input into action, and how collective behavior may naturally arise from this navigational circuit.

We also would like to mention that, the head direction literature aims to address how animals can form an allocentric representation of their head direction (using mechanisms such as path integration in combination with the utilization of external cues). Given such a representation, animals can also transform an egocentric representation of bearings into an allocentric one. We do not aim to address this capacity (which has been subject to many studies in the head direction literature, is subject to variation across the levels of organization, and is poorly understood in the vast majority of species). Rather, we aim to reveal its consequences for individual and collective behavior. However, head direction and heading direction are strongly correlated in many animals. Please also see Methods, lines 625 to 659, especially, 631 to 639:

We can think of the animal as always turning towards the direction that it is moving to (heading direction), such that the animal instantaneously updates its head to coincide with its heading direction (see Fig. 1C). We note that this assumption originates from the parsimony of our model and its focus on addressing how animals establish a “goal direction” (rather than, e.g., focusing on how animals maintain an allocentric head direction with respect to landmarks [87]), making the reasonable assumption that they can steer towards their goal [41, 44, 53, 77]. Furthermore, biologically, this intu-

ition is supported by empirical data based on which head and heading direction coincide (in animals with rigid bodies such as fish [56]) or are strongly correlated [41, 54, 55, 55, 68, 69]).

Another point that we would like to clarify here, is the definition of allocentric and egocentric representation. As this issue is related to comments 1 to 3, and it appears to us has originated from our failure to clearly articulate the differences between our work and the head direction literature in the previous version, and as disambiguating it helps us to better respond to some of the other comments, we provide this disambiguation here.

The issues raised by the referee also made us realize ambiguities in our explanation of the model with respect to the use of “egocentric” and “allocentric”. Consequently, to improve clarity, we have modified our description of the conceptual basis of the model to make clear that we do not imply that organisms differ in how they represent their positions in space (in the present work this is always egocentric, i.e., bearings originate from the animal’s position). Thus the models only differ with respect to animals’ representation of bearings (angles) towards objects in space.

Originally we had considered these angles to be encoded either:

A) with respect to the direction of movement (heading) of the animal — the egocentric model.

B) with respect to one or more external cues, such as a stable landmark, magnetic field, polarised light etc. — the allocentric model.

In the latter case, the representation of bearings towards objects does not require an animal to know its own heading direction since bearings can be computed as angles with respect to an external cue (although, biologically, many animals may utilize information about their own heading to compute, or help computing, an allocentric bearing). As mentioned before, we do not aim to address how an allocentric representation of space is achieved (which, loosely speaking, is the aim of the head direction literature). Rather, by assuming such a capacity exists, we explore its consequences.

To aid in this clarification we have also modified Fig. 1. The new text reads as (lines 56 to 70):

Maintaining a compass does not imply that each individual knows which way is north, or that different individuals share a common allocentric frame of reference; indeed, animals must typically re-tether their compass as they move through space and contemporaneous salient cues, e.g., visual [55] or magnetic cues [61, 62, 73], change. It only means that individuals can use available sensory information to maintain egocentric bearings, such as towards objects, as well as (thanks to their compass) to have the capacity to transform egocentric representations to allocentric representations on their ring attractor networks [41, 48, 74]. Therefore, while all bearings we consider here are egocentric in terms of their point of origin—centered on the animal—their bearings can be encoded in an egocentric and/or an allocentric (polar) reference frame in the brain [48, 55] (see Fig. ??). Importantly, this does not imply the existence of a cognitive map or absolute knowledge of object locations in Cartesian space (e.g., knowing a tree’s coordinates as (X, Y) independently of the individual’s location [75, 76]). Rather, we refer to the encoding of bearings in egocentric or allocentric terms [48]. Future work could extend this framework to incorporate more complex spatial representations, such as the Cartesian encoding observed in mammalian brains [50, 75, 76], but here we focus on the simpler mechanisms that may underpin the evolutionary origins of collective motion in invertebrates and vertebrates.

We further clarify the definition of allocentric and egocentric in response to comment 3.

Reviewer N. 1:

2. The formulation of the model needs improvements

I found the formulation of the model to be a bit problematic and not entirely satisfying. Currently,

the model is specified by the Hamiltonian, and Glauber dynamic is used to simulate the network dynamics. The influence of the external sensory cues was attached to the network in a rather post-hoc way.

Thus, this model is not directly constructed based on a group of neurons and the connectivity. I am concerned that the simulated dynamics may not faithfully reflect the network dynamics of a more realistic network model.

The paper would be strengthened if a fully-consistent and realistic neural network model could be constructed and simulated, by coupling a set of ring attractor network modules studied in neuroscience, such as the classic model by Zhang (1996). Note that in the Zhang model (as well as many other versions of the ring attractor models), one can explicitly write down the dynamical equations governing the dynamics of individual neurons in terms of membrane potential and firing rate. Neurons encoding sensory inputs can be wired together with the neurons on the ring by defining specific connectivity or through some learning rules (e.g., Hebbian plasticity).

Zhang, Kechen. "Representation of spatial orientation by the intrinsic dynamics of the head-direction cell ensemble: a theory." *Journal of Neuroscience* 16.6 (1996): 2112-2126.

Response:

We are thankful to the reviewer for this insightful remark which helped us to significantly improve our work by introducing a new model better aligning the referee's suggestion (the *neural field model* presented in 4.1.2), which is thoroughly analyzed in the revised manuscript and its Supplementary Information. In addition, in the Supplementary Note 1, we have provided a general conceptual framework.

In addressing this comment, we have referred to the foundational formulation of Amari, which has been employed by many studies, including the one mentioned by the reviewer, to explain a wealth of phenomena ranging from sensory processing to motor movement, and head direction cells (Please see the discussion in Section 2, and Methods in the revision).

The reason we found it is best to refer to a more foundational formulation is the aim of our work. As we have clarified above, we do not aim to model how an allocentric or ego-centric perception of space is formed. Rather, we aimed to address how (individual and collective) decisions are made by establishing a goal direction. Given that the main aim of our work is to understand individual and collective behavior, we believe this goal is fully reached by making the model as simple as possible to achieve, arguably, the most parsimonious model that can account for collective behavior. We noted that this goal can be achieved by a quite simple one-layer Amari-style network. Thus, despite providing a general extendable *conceptual* framework (in Supplementary Note 1), we have focused on such a parsimonious model. However, we note that the new model, while providing the biological plausibility that the reviewer hinted at, still differs from the head direction literature (e.g., Zhang 1996), in that, it has a different aim and scope, as we have clarified above. Below, we summarize why we believe the parsimony of our model is an important advantage (not a limitation) that we retain in the improvement of our model.

Most notably, our aim is not to provide an understanding of how animals form an allocentric representation of space, which, as noted before, has been subject to many studies. Rather, we aim to provide an understanding of their consequences for individual and collective behavior. Lines 78 to 82:

Here, we focus our attention on this less-explored aspect of decision-making and make the reasonable assumption that animals can turn towards their goals. Thus we do not explicitly model how animals maintain their allocentric heading, which they are known to be able to do, but rather how sensory information—with a specific focus on visual information—may be integrated to create time-varying goal directions.

Besides, the variation among species in neural architecture and mechanisms using which an allocentric representation of space is formed, can provide important advantages for a parsimonious model, which due to its parsimony, can be applicable to a wide array of species. Please see, e.g., lines 38 to 39:

*While neurobiological details differ among species, a ubiquitous motif for encoding angular information in both the invertebrate [54, 55] and vertebrate [44, 56–59] brain, are “ring attractor networks”. A ring attractor network is a recurrent neural circuit in which localized excitation and long-range inhibition maintain a “bump” of electrical activity, with recurrent excitation maintaining the bump even in the absence of sensory input. Ring attractors can have multiple inputs, often from other ring attractors and/or from sensory modalities. Their functional ring-like topology (which in some cases, such as the ellipsoid body of the fruit fly *Drosophila*, is literally also a morphological ring [54]), makes them ideal structures for the integration and representation of angular information.*

Multiple interconnected and intercommunicating ring attractor networks coexist in the brain. Central to spatial navigation is animals’ neural “compass”, often termed their “heading compass” or “head compass”, in which the cellular activity rotates as the animal changes heading, allowing estimation of body/head orientation relative to visual [60] (and in some species also magnetic [61–63]) cues. Prominent visual cues employed to “tether” the compass include polarized light [64–66], the sun [60, 61] and prominent distant, and therefore relatively stationary, objects in the environment [61, 62]. In this way, the animal can maintain an allocentric reference frame for its heading, i.e., its orientation with respect to external cues [43, 48, 55, 67–72]. While some species, such as fish [56], have rigid bodies, in others, such as mammals, animals can maintain head direction in addition to heading [57, 58]. We note that head and heading directions have yet to be disambiguated in insects [54, 55].

Finally, we note that while our work is inspired by biological networks, we note its significance in the realm of artificial neural networks (lines 603 to 605):

Besides, while our work is inspired by biological neural networks, it can potentially integrate biological and artificial neural networks by motivating new areas of research in artificial neural networks, such as swarm robotics.

In summary, we sincerely thank the reviewer for their insightful comment, which has motivated and helped us to significantly improve our work by including a neural field model. The new model is added to the method sections, and its phenomenology is extensively studied in the Results and the Supplementary Information. These steps significantly improve biological reality, mathematical elegance, scalability, generalizability, future development, and the robustness of our work.

Reviewer N. 1:

3. Issues with allocentric v.s. egocentric representations

A central aspect of the work is the comparison between models based on the allocentric v.s. egocentric representation. There are several concerns related to this point that I hope the authors could address.

- First, how the egocentric-based and allocentric-based representations were modeled needs a better explanation. The descriptions in the Method section on page 11 and 12 are somewhat ambiguous. The main difficulty is the model is described by a hybrid of quantitative and qualitative components. It would be helpful if the authors could define the full model by writing down the equations for the allocentric version of the model, then do the same thing for the egocentric version of the model.

Response:

Thank you for this question. In addressing this question, we made extensive changes as follows: A new

panel is added to Figure 1 to illustrate the definition of allocentric and egocentric reference frame (Fig 1.C), a better discussion and mathematical definitions are added to the model definition in Methods (4.1.1 and 4.1.2).

Regarding the definition of allocentric and egocentric, we have significantly improved the models presentation and better explained the definition of allocentric and egocentric representation. All the mathematical and dynamical equations are the same for the allocentric and egocentric versions of the model. The only difference is the definition of the reference frame using which the network represents objects. While in both cases the reference frame originates in the individual, in the allocentric version, this reference frame is world-centric, i.e., a frame “anchored” in the environment, such that it does not rotate as the animal body axis (heading direction) rotates in space. In the egocentric version, this frame is attached to the agent both translationally and rotationally. That is, as the agent moves in space, the representation of targets also changes (the egocentric angular representation of targets rotates accordingly). The distinction between these two definitions of the reference frames is also better illustrated in a new cartoon added to Figure 1 (panel C). Below is the text in the revised methods where allocentric and egocentric are detailed (lines 625 to 659):

The key question, where the agent’s representation of space plays a role, is how to specify the angles, $\hat{\alpha}_i$. With an egocentric representation of space, the angles $\hat{\alpha}_i$ encode for a polar direction in the body-centered coordinate of the animal. Thus, we can take, $\hat{\alpha}_i = \alpha_i$. This is implemented by indexing neuron with respect to an arbitrary position of the animal, e.g., head, such that neuron $i = 1$ with angle $\alpha_1 = 0$ represents the heading direction of the animal (assuming the animal’s head is aligned with the direction that it is heading to), and neuron i receives input from an angle centered on $\alpha_i = \frac{i-1}{N}2\pi$ with respect to the animal’s heading. We can think of the animal as always turning towards the direction that it is moving to (heading direction), such that the animal instantaneously updates its head to coincide with its heading direction (see Fig. 1C). We note that this assumption originates from the parsimony of our model and its focus on addressing how animals establish a “goal direction” (rather than, e.g., focusing on how animals maintain an allocentric head direction with respect to landmarks [87]), making the reasonable assumption that they can steer towards their goal [41, 44, 53, 77]. Furthermore, biologically, this intuition is supported by empirical data based on which head and heading direction coincide (in animals with rigid bodies such as fish [56]) or are strongly correlated [41, 54, 55, 55, 68, 69]).

In the allocentric version of our model, instead, the direction for which neurons code is independent of the agent’s heading direction (where it is moving to) or bodily coordinate (how it is posed or which direction it is facing). Rather, the neurons code for a direction in a world-centric polar coordinate. Thus, the center of the receptive field of neuron i is an angle $\hat{\alpha}_i = \frac{i-1}{N}2\pi$ with respect to an absolute reference frame independent of animal’s orientation, e.g., a world-centric east (positive x -axis). This can be achieved, for instance, by anchoring to one or more external cues, such that, as the agent moves in space, the ring attractor network does not rotate with the agents’ body axis (see Fig. 1C). Clearly, this does not mean that all the agents necessarily share common allocentric reference frames; how the agents define the zero of their coordinate (and thus how they define, e.g., north) is a matter of indexing the neurons and is inconsequential for their neural dynamics and its resulting movement pattern. Thus, “anchoring” to different external cues (or not having a consensus on which direction is north), does not affect the collective movement of the agents.

To further clarify how an allocentric representation of space can be achieved, we can write, $\vec{\alpha}^{\text{allo}} = \vec{\alpha}^{\text{ego}} + H^{\text{allo}}$, where H^{allo} is the agent’s heading direction in an external (allocentric) reference frame (an external polar reference frame not to be confused with the agent’s allocentric reference frame). It is known that animals can maintain such an allocentric representation of their heading or head direction using ring attractor networks (for instance, via path integration combined with the

utilization of environmental cues), which they utilize to maintain an allocentric representation of space [48, 55–58, 70]. To do so, in our allocentric model, we have assumed the agent has such a capacity and encodes polar directions in an allocentric way.

In addition, we have provided a modified model where agents move with constant speed (like classical models of collective behavior, e.g., Vicsek or Couzin Models), but their heading direction is computed from the ring (lines 720 to 729). In this part, we have provided the mathematical equations determining the agent’s heading, which can be taken to be the zero of the egocentric reference frame. In the allocentric case, the zero of the reference frame is independent of the agent’s heading.

Reviewer N. 1:

- Second, there is substantial evidence suggesting that the brain maintains multiple reference frames. It may well be the case that the brain generate representations of the environment both in allocentric sense and in egocentric sense simultaneously. In the authors’ framework, it felt that one was forced to choose one or another. How can the results presented in the paper reconcile with the coexistence of multiple reference frames?

- Third, the authors seemed to consider the allocentric reference frame as being fixed. What if the neural system re-anchors to a different sensory cue? (Re-anchoring is not uncommon.)

Response:

Thank you for these fruitful comments.

To address these two points, we have taken a number of steps, from better clarification to introducing new concepts, methods, model extensions, and results, as detailed below. We begin by addressing the third point (as it seems to be more elementary).

Regarding anchoring or re-anchoring, we have provided discussions (Method Section, Modeling framework, 4.1) and conceptual clarifications that while the mechanisms using which organisms form an allocentric perception of space are not fully understood, a combination of using external cues and path integration seem to play a role in many organisms. We thus used “anchoring”, as how the organisms define their allocentric reference frame (by supposedly, anchoring it to one or more environmental cues.) In this sense where and how to anchor, e.g., whether all individuals should have the same reference frame, or different reference frames, or how to define the zero of the reference frame (e.g., East) do not affect the simulations. While intuitively clear, to also implement that, in all the new simulations added to the revision and SI, the zero of the allocentric reference frame of individuals is randomly assigned.

These points are clarified, for instance in the model definition in the Methods (lines 625 to 659 quoted above) and also in the introduction (lines 56 to 62):

Maintaining a compass does not imply that each individual knows which way is north, or that different individuals share a common allocentric frame of reference; indeed, animals must typically re-tether their compass as they move through space and contemporaneous salient cues, e.g., visual [55] or magnetic cues [61, 62, 73], change. It only means that individuals can use available sensory information to maintain egocentric bearings, such as towards objects, as well as (thanks to their compass) to have the capacity to transform egocentric representations to allocentric representations on their ring attractor networks [41, 48, 74].

Regarding coexistence, based on this constructive comment, we have extended our framework to address the switch (or coexistence via rapid switching) between allocentric and egocentric reference frames (4.1.3 and 2.6). This extension of our work can readily reveals some of the high potentials and scalability of our work. Further exploration of this topic is discussed as a future direction in

the Discussion. This addition quantitatively addresses the reviewer’s question, and confirms that the model can be straightforwardly extended to more complex scenarios and important insights can be reached.

In the switching scenario discussed here, individuals randomly switch between allocentric and egocentric reference frames in a continuous way, meaning that when they switch between an allocentric and egocentric reference frame they can redefine their allocentric reference frame via “reanchoring”. We have shown that this simple scenario already provides important insights.

This topic is introduced in the Methods (lines 712 to 718) and extensively analysed in the Results (2.6, lines 474 to 529) and the Supplementary Information.

We have also modified the abstract, Introduction (lines 105 to 107),

Although a purely egocentric encoding fails to produce collective motion, rapid alternations between allocentric and egocentric frames can enhance global order.

And Discussion (lines 584 to 591):

We also considered the fact that animals can employ both egocentric and allocentric representations of space, with the ability to integrate and/or transition between them (e.g. rapid resets to landmarks [46, 48, 48, 50], temporal switching [99, 100], or coexistence of both via parallel information processing in different brain regions [45, 47]). Using a minimal random switching scheme, we find that rapid, intermittent flips between frames (with their attendant “re-anchoring”) can enhance global alignment beyond the pure-allocentric case. Whether, and if so how, animals schedule their frame switches, as well as implementing more sophisticated context-dependent switching and/or integration of such representations, is a promising avenue for future work.

Reviewer N. 1:

- Fourth, the results based on the comparison of the egocentric-model and allocentric-model paint a pretty complex picture. My understanding of the results is that allocentric representation only convey an advantage in certain regimes. For example, in section “Finding and staying close to a static target”, the egocentric representation in fact can have an advantage. The Abstract should be revised to more accurately reflect these results.

Response:

Regarding this point, as the reviewer notes, in the spin system model, egocentric representation performs better in finding and staying close to a static target. However, the new neural field model suggests that allocentric and egocentric perform very closely in such context (in some parameter regions, 2.3 and SI). Thus, this claim is supported only partially by the evidence provided by our study. However, while we have discussed this in the article (Results and Discussion), due to a more comprehensive revision of the abstract (based on which we have focused on the most important findings of our study, given the word limit) the current abstract does not discuss the findings regarding individual motion and information acquisition.

Reviewer N. 1:

4. The assumption on the relationship between the movement and neural activity needs justification.

One crucial assumption of the models proposed was that “the agent’s movement is dictated by the activity of its neural network, where the velocity vector is computed based on the activity of the neurons. ” It is not obvious to me why this is a good assumption. In previous ring attractor networks, the activity bump on the ring typically encodes the current stimulus, rather than the decision or movement. In particular, this is the case for the head direction system. Any experimental evidence

that the authors could provide? In any case, it would be helpful if this important assumption can be justified or discussed in more detail.

Response:

As mentioned above, the aim and scope of our work are different from previous work which has used ring attractor networks to understand head direction systems. In this regard, our work aligns with some recent works cited in the manuscript (Sridhar et. al. 2021), which have used ring attractor networks in this way (to represent agents' goal vector rather than head direction). Recent empirical evidence has also supported the phenomenology suggested by such models, showing that the phenomenology of animal decision-making can be explained by such models.

Here is the relevant text from the Introduction (lines 71 to 90). We note that this text is also used to better clarify the differences in the aims and scope of our article and the head direction literature:

*In addition to their heading compass, animals have also been found to encode their “goal direction” in a ring attractor network. Here, the bump of activity represents the desired direction of travel for the animal at the present moment in time [41, 43, 44, 53]. While it is known that animals turn towards their goal vector during navigation, with the neural circuitry responsible for converting allocentric goals into appropriate egocentric steering controls having been dissected in *Drosophila* [41, 53], relatively little work has been yet conducted into how the goal vector is itself established when there are multiple alternatives [77].*

Here, we focus our attention on this less-explored aspect of decision-making and make the reasonable assumption that animals can turn towards their goals. Thus we do not explicitly model how animals maintain their allocentric heading, which they are known to be able to do, but rather how sensory information—with a specific focus on visual information—may be integrated to create time-varying goal directions.

Our use of a ring attractor network to explore decision-making with respect to establishing a goal direction is motivated by its neurobiological plausibility [43], and that we previously found that a ring attractor model could accurately predict the time-varying directional movement decisions exhibited by individual fruit flies, locusts, and zebrafish, in scenarios involving two or more discrete static (fruit flies and locusts) and moving (fruit flies, locusts and zebrafish) options [40, 78, 79]. In this work, we were, however, unable to account for how collective motion emerges in animal groups. Importantly, similar to all previous models of collective behaviour, in our previous ring attractor models [78, 79], we had assumed that animals employ an egocentric representation of space.

Reviewer N. 1:

5. The functional importance of attractor dynamics needs to be further clarified.

The authors motivated the ring-attractor model based on the literature of the ring-attractors in the fly and rodent head direction system. This makes sense. But is the continuous dynamics actually important for the results in the paper?

The paper would be strengthened if the authors can construct alternative models that are similar to these studied in the paper, but without attractor dynamics. Right now, the role of attractor dynamics is unclear.

It is possible that attractor dynamics is not essential. If that's indeed the case, it would lead to a different interpretation of the results, and likely also a simplified account of these results (perhaps a model without a neural circuit architecture would be sufficient to explain the results)? Looking into these issues would help distill the key factors in the model that leads to the key predictions.

Response:

Thank you for this insightful comment. Based on this comment, we have added new modeling and analysis to the parameter dependence Section, 2.7, and SI. We have also provided an SI Video (SV.17).

To address this comment, we started by developing such alternative models. But then realized such alternative models are nothing but our very own models in which the recurrent connections are removed by setting the synaptic connectivity to zero. The resulting models are identical to our models in every aspect (including an allocentric or egocentric perception of space), but lack recurrent connections. The results show that removing recurrent connections leads to a trivial phenomenology. As intuitively expected, the collective collapses into each other due to social attraction. The detailed phenomenology is studied in the SI and a SI video is provided. Thus, we conclude that recurrent connections are essential for both collective movement and its complexity.

Reviewer N. 1:

6. Experimental relevance of these results remains unclear

The connection to experimental results is only briefly mentioned in the paper. To understand the implication and significance of these modeling results, a more direct and systematic comparison to the experimental literature would help a lot. I am not sure if quantitative comparison is possible in the case. But a more careful treatment of when the models generate consistent or inconsistent predictions with the data would benefit the paper.

Response:

We thank the reviewer for this comment. As the reviewer mentions, this is beyond the scope of this work given its theoretical focus and concentration of findings. However, a better discussion of empirical evidence is added in the revised manuscript both in individuals: lines 551 to 556:

At the level of individual navigation, our models suggest an egocentric representation of space may provide advantages in navigating in relatively stationary environments and towards nearby objects. These findings are consistent with empirical observations that some organisms tend to represent nearby objects more in an egocentric, and those far away more in an allocentric way [100]. On the other hand, our models predict that allocentric representations facilitate the pursuit of moving targets. This seems to be consistent with empirical findings in both insects [41] and mammals [44].

And collectives, clarifying how our work aligns empirical evidence better than alternative approaches (lines 569 to 576):

While alignment is observed in many species—such as starlings [101, 102] and shoaling fish [38, 39]—our work suggests it may be an emergent by-product of allocentric (or coexistence of allocentric and egocentric) representation of space by animals— a view supported by studies that failed to find empirical evidence for explicit alignment among fish [38, 39] or swarm-forming locusts [40]. Our work shows that local alignment can arise as a form of consensus dynamic (not dissimilar, conceptually, to models of collective information acquisition [104]) for agents who have an allocentric representation of bearings (their own heading direction, and the bearings towards others, are within a world-centered frame).

In addition, in several places, we have added references to, and better discussions of, empirical and neurobiological literature, underlying our model construction and its phenomenology and predictions (e.g., allocentric and egocentric perceptions and switching between the two in the Results and Methods and “goal direction” in the Introduction). Together with the references to experimental literature in the previous version, we hope this more systematic treatment of the biological literature makes the empirical relevance of our allocentric flocking framework much clearer.

Reviewer N. 1:

***** Other concerns/suggestions for improvements:

** In the Introduction, it is stated that “it has recently been shown that ring attractor networks also account for how fruit flies, locusts and zebrafish (and thus both invertebrates and a vertebrate) make decisions when choosing between multiple spatial “targets”...” The references cited there were modeling studies. Is there any neuroscience evidence suggesting that the ring attractors actually encode the decision of head turning? My understanding of the literature is that head direction cells in the fly and rodent are considered to encode the current head direction, not the turning decision. But I could be missing something. (This is related to Point # 4 above)

Response:

In light of the improvements in the modeling framework, and to better motivate our work and its contribution, this part is removed (and revised) in the revision. However, two lines of work exist. Ring attractor networks have been also found to be utilized in animals to form a “goal direction”, and it is shown that they are involved in animals’ capacity to steer towards their goal. This literature is very recent neurobiological evidence (but to our knowledge modeling work is still lacking). This discussion can be found in lines 46 to 77, particularly:

In addition to their heading compass, animals have also been found to encode their “goal direction” in a ring attractor network. Here, the bump of activity represents the desired direction of travel for the animal at the present moment in time [41, 43, 44, 53]. While it is known that animals turn towards their goal vector during navigation, with the neural circuitry responsible for converting allocentric goals into appropriate egocentric steering controls having been dissected in Drosophila [41, 53], relatively little work has been yet conducted into how the goal vector is itself established when there are multiple alternatives [77].

Another line of evidence is some very recent works that show that ring attractor networks can explain the *phenomenology* animal decision-making. This is clarified in lines 83 to 90:

Our use of a ring attractor network to explore decision-making with respect to establishing a goal direction is motivated by its neurobiological plausibility [43], and that we previously found that a ring attractor model could accurately predict the time-varying directional movement decisions exhibited by individual fruit flies, locusts, and zebrafish, in scenarios involving two or more discrete static (fruit flies and locusts) and moving (fruit flies, locusts and zebrafish) options [40, 78, 79]. In this work, we were, however, unable to account for how collective motion emerges in animal groups. Importantly, similar to all previous models of collective behaviour, in our previous ring attractor models [78, 79], we had assumed that animals employ an egocentric representation of space.

Reviewer N. 1:

** I felt the flow in the Results section could be improved. It was a bit slow to get to the key results. Perhaps combining and compressing the subsections “Spontaneous neural dynamics” and “From spontaneous neural dynamics to individual motion ” would help.

Response:

We have implemented this suggestion.

Reviewer N. 1:

**It would be useful to label/number the subsections in the Results section.

Response:

We have numbered all the subsections in the revision.

Reviewer N. 1:

** How does the connectivity pattern affect the results? For example, if varying the width of the

connectivity profile, would that affect the results?

Response:

The parameter dependence is extensively studied in the SI for all the parameters and both models. For instance, the robustness of the results for different values of the width of the connectivity (tuning parameter ν) is shown. While in all the cases our core conclusion, including the emergence of local and global alignment holds, the emerging patterns depend on the parameters (and model formulation, given in the revised version several extra models are presented), possibly in non-monotonic ways (for instance in the simulation presented in the SI we observe smaller ν may lead to more alignment, but we can not exclude that this trend depends on other parameter values. Besides, the two models may provide different insights). Given the rich suite of motion patterns produced by the model, many of which seem to have counterparts in biological flocks, a more extensive study of the patterns observed for different parameter values (or, for instance, beyond the confine of cosine-shaped connectivity patterns) and comparison with empirical patterns could be subject to several future studies.

Reviewer N. 1:

** In the Method section, it would help if the equations for defining VOP and geometric VOP could be included.

Response:

They are included in the revision.

Reviewer N. 1:

***** Suggestions for the Figures:

I also have a list of suggestions regarding the presentation of the figures/figure legends.

Fig. 1B: I think the cartoons for the two insects meant two different individuals. If that is the case, it would be useful to label them. Otherwise, some may also interpret it as an insect moves to a different spot with the head direction fixed.

Response:

We have labeled the individuals in the revision.

Reviewer N. 1:

Fig. 2: in the second and third columns, please plot and label the axes. What spaces were these plots based on? 2-D physical space? This may not be as obvious. Also, please clarify the meaning of “200 units”. What is the unit?

Response:

Yes, it is 2-D space. Labels and axes are plotted in the revision. The scale bar simply was meant to show the spatial scale. “Unit” is an arbitrary measure of (dimensionless) distance. Because this is simulation data, a standard unit (such as a meter or centimeter) does not exist and it could have alternatively left blank.

Reviewer N. 1:

Fig. 3G,H,I: please define and explain the quantifies (on the y-axis) being plotted in the figure legend/caption.

Response:

This is corrected in the revised figure caption. (Further details on the simulation setup are provided in Statistics and Reproducibility in Methods to keep the caption short enough by limiting to the essential information.)

Reviewer N. 1:

Fig. 4: It would be useful to add (sub-)titles in individual figure panels to indicate the allocentric v.s. egocentric cases.

Response:

This suggestion is implemented in the revision.

Reviewer N. 1:

Fig. 5: This figure is difficult to interpret on its own without going back and forth together with the text. Please improve the figure legend and explain what was plotted. In particular, it would be useful to remind the readers the meaning of the variables being plotted.

Response:

The caption is updated to better clarify the message of the figure.

Reviewer N. 1:

Fig. 8: In the legend, it is stated that “B and C: The motion pattern (A) and the neural activity (B) of a subgroups of 3 coherently moving agents among the 80 agents presented in A are shown. ” This is confusing. How is Fig 8B showing the neural activity? I thought that should be panel C and E. I think there might be an error here.

Response:

We thank the reviewer for noticing this typo. The error is corrected in the revision.

Reviewer N. 1:

Fig. 9: would it be possible for the authors to add another set of curves for $L=100$? Having results based on parameters that spans several orders of magnitude would make the argument more convincing.

Response:

This figure is removed in the revision to make space for two new figures and give a better structure to the manuscript. Besides, as the message of this figure, lack of density dependence phase transitions is more extensively studied in figures 6 and 7, it was not deemed essential, and more extensive analysis of parameter dependence is provided in the Supplementary Information (including the same figure panels).

3 Response to Reviewer N. 2

Reviewer N. 2:

Peer Review for "Allocentric Flocking" by Salahshour & Couzin

Summary

This paper introduces a compelling theoretical model of collective behavior based on ring attractor neural networks, comparing allocentric (world-centered) and egocentric (self-centered) spatial representations. The authors demonstrate that allocentric representations can generate coherent, large-scale flocking behaviors, while egocentric representations primarily produce static aggregations. Their model derives from Hamiltonian energy minimization principles and is explored through systematic simulations across various noise levels and social coupling strengths.

Strengths

- The paper is technically sound with clean model implementation and clear assumptions.
- It represents an excellent contribution to the field of physics-inspired collective behavior models.
- The simulations and phase space analysis are meticulously executed, with results that appear robust across different parameter regimes.
- The effort to ground flocking behavior in neurobiological ring attractor mechanisms provides a fresh and intriguing perspective.

Response:

Thank you for your careful reading of the manuscript, for finding the work of interest, and for their constructive suggestions which we have used to significantly improve the manuscript.

Reviewer N. 2:

Major Comments

1. Neuroscience Terminology and Framing

The paper uses several terms from cognitive neuroscience in ways that might not align with their conventional meanings:

- "Belief updating" typically suggests probabilistic or Bayesian inference processes, whereas here it refers to deterministic shifts in attractor network activity. I'd recommend clarifying this usage or selecting alternative terminology.
- Similarly, "internal model" and "world model" appear without distinguishing between basic embodied representations (as found in insects) and sophisticated predictive models in complex organisms. This distinction would strengthen the paper's theoretical foundations.
- The supporting references for "internal models" (e.g., [22-25]) could be better aligned with the established neuroscience literature.

Response:

We have improved the neuroscience terminology and the discussion of related literature. We have replaced "belief updating" with terminology that does not cause confusion, and in the only place where the internal model is used, we have added the new citations, better aligning our usage of this phrase to represent the general background in collective movement research (line 18).

Reviewer N. 2:

2. Suggested additions for consideration:

- For neural "world models": Keller & Mrosovsky (2018, *Neuron*), Pezzulo et al. (2017, *TiCS*)
- For computational world models: Ha & Schmidhuber (2018), Lehman et al. (2022)

Response:

Thank you for pointing out this useful literature, which we have added to improve our citation to literature.

Reviewer N. 2:

3. Model Construction and Emergent Properties

While elegant, the model seems somewhat constructed to yield its primary finding—that allocentric representations produce stable flocking while egocentric ones don't. The ring attractor with energy minimization and specified input fields makes this outcome somewhat expected. I would appreciate a more explicit acknowledgment that the observed behaviors are substantially encoded in the system's structure rather than emerging unexpectedly. This doesn't diminish the work's value but would position it more accurately within the field. A discussion comparing this framework to other Ising-like models in neuroscience would help highlight what's genuinely novel about the navigational coupling introduced here.

Response:

Thank you for your positive view of our work.

We first provide our response to the question regarding a comparison of our model compared to other similar models. Before providing our detailed response, we note that in the revised manuscript we have added a new model (which we have called the neural field model) that generally reproduces the results of our spin system model and provides additional interesting phenomenology. The two models can be considered complementary models. This new model significantly improves the novelty, robustness, biological realism, and appeal of our work to neuroscience and AI literature.

Regarding the novelty of the spin system model, we have referenced the most related use of spin system models in neuroscience starting from Hopfield's work and mentioned the most similar works to our *spin system model*, which are a series of recent studies [e.g., Sridhar et al. 2021], which aimed to model individual decision-making in the presence of conflicting preferences. While our spin system model uses similar tools, it has a starkly different aim and scope.

We have clarified that the allocentric and egocentric representation of space, and using this approach to ground collective behavior on navigational circuits of animals, have not been considered in previous models. Despite using similar tools, our model differs from previous similar models [e.g., Sridhar et. al. 2021] in key ways, such as differences in the implementation of egocentric representation, and lack of allocentric representation in previous works, and hard-coded affinity of the agent to the target which makes these models unsuitable for studying collective behavior. Given these stark differences, the similarity of our spin system model with previous models is mainly conceptual (besides using similar mathematical tools) and relies on an attempt to utilize a ring attractor network to address animal decision-making, which in our revised manuscript is better articulated as addressing the question that how animals establish a “goal direction”.

Regarding the findings of the work and their significance, we guess that this comment may originate from a misunderstanding that in the allocentric model collective motion arises due to a shared consensus, we would like to clarify that our results are independent of the agents’ reference frame’s origin (zero). That is, the agents do not necessarily have a consensus on which way is, e.g., North. Thus, collective motion in our model does not arise from shared information (of a common direction).

Having this point in mind, we would like to distinguish two findings:

- Firstly, that allocentric representation of space produces collective movement without additional rules of interaction. We believe that this result is difficult to predict given the model construction (and in fact, at first, the model was not constructed to yield this outcome and only later we realized we could use the model to formulate a new approach to understand collective behavior). In contrast to traditional, rule-based models (e.g., Vicsek and Couzin Models), where collective motion emerges out of model assumptions (e.g., explicit local alignment), such properties are emergent in our model.
- The second finding is that (while an allocentric representation of space does) an egocentric representation of space does not produce collective motion. We fully agree that this result makes intuitive sense a posteriori, both mathematically and biologically. However, as simple as this conclusion may seem, it was not clear to us a priori and before observing the model phenomenology. We agree with the reviewer and are thankful to them for noticing that, this fact “doesn’t diminish the work’s value”, but may even better show the value of the work by clarifying as simple and intuitive as one of its key findings was, it has not been noted before in the literature.
- Furthermore, in the revision, surprisingly, we have shown that switching between an allocentric and an egocentric representation of space can improve highly ordered collective motion (2.6), which while makes sense a posteriori, may better reveal unexpected insights that come out of our approach (given that an egocentric representation by itself does not even produce collective motion).
- Finally, the new model added in the revision, by reproducing all the overall phenomenology observed in the previous model, but from a different modeling perspective, can increase both the robustness and generality of our approach and better show that our results are independent of the particular ways the model is constructed (such as energy minimization in neural dynamics).

Reviewer N. 2:

4. Treatment of Egocentric Representations

The manuscript’s conclusion that egocentric representation fails to support collective motion risks overgeneralization. In biological systems, egocentric representations work alongside velocity integration, memory, and goal-directed behaviors—elements absent from this model. I’d be interested in the authors’ thoughts on what mechanisms might enable egocentric flocking in real systems, such as short-term memory, learned movement patterns, or heading stabilization.

Response:

Thank you for this insightful remark. That in real animals, even simple organisms such as fruit flies, egocentric representation alone can not be found is completely true and one of the points that we clarify, and even capitalize on in our manuscript in several places. The reviewer’s observation that a bare and simple egocentric representation of space is not found in animals strengthens our argument: all the realistic additional pathways that accompany an egocentric representation of space in animals can be among the pathways that different organisms use to form an allocentric representation.

In the revision, we clarify this point in several places. For instance, please see lines 56 to 64 in the revision:

Maintaining a compass does not imply that each individual knows which way is north, or that different individuals share a common allocentric frame of reference; indeed, animals must typically re-tether their compass as they move through space and contemporaneous salient cues, e.g., visual [55] or magnetic cues [61, 62, 73], change. It only means that individuals can use available sensory information to maintain egocentric bearings, such as towards objects, as well as (thanks to their compass) to have the capacity to transform egocentric representations to allocentric representations on their ring attractor networks [41, 48, 74]. Therefore, while all bearings we consider here are egocentric in terms of their point of origin—centered on the animal—their bearings can be encoded in an egocentric and/or an allocentric (polar) reference frame in the brain [48, 55]

And also, lines 652 to 659 in the Methods:

To further clarify how an allocentric representation of space can be achieved, we can write, $\vec{\alpha}^{\text{allo}} = \vec{\alpha}^{\text{ego}} + H^{\text{allo}}$, where H^{allo} is the agent’s heading direction in an external (allocentric) reference frame (an external polar reference frame not to be confused with the agent’s allocentric reference frame). It is known that animals can maintain such an allocentric representation of their heading or head direction using ring attractor networks (for instance, via path integration combined with the utilization of environmental cues), which they utilize to maintain an allocentric representation of space [48, 55–58, 70]. To do so, in our allocentric model, we have assumed the agent has such a capacity and encodes polar directions in an allocentric way.

Besides, in the revision, we note that allocentric and egocentric representations coexist in animals and provide extensions of our model to study this realistic phenomenon and reveal important insights on how coexistence can improve alignment (2.6). Furthermore, we discuss some of the mechanisms closely related to those mentioned by the referee, to clarify how coexistence of allocentric and egocentric representation may be achieved: lines 712 to 718

In the extension of our model to consider switching between reference frames, we assume the agent can continuously switch between the two [allocentric and egocentric] reference frames, such that it does not temporarily lose its sense of orientation when switching. ... Biologically, such an ability to change the allocentric reference frame, which we refer to as “re-anchoring”, can result from maintaining a sense of orientation via exploiting external cues [48], path integration [48, 50], or memory-based cues [46]

Regarding what mechanisms may enable “egocentric flocking”, as the reviewer notes, it is intuitively expected that goal-oriented behavior, in the sense that all the agents have similar preference for a target, can lead to some sort of stimuli-induced collective movement toward the same direction. However, the true challenge in the context of collective motion is to explain the ubiquitous and non-trivial form of self-organized flocking that occurs without such apparent shared affinity to environmental stimuli or a destination (as we have detailed in response to your comment 3).

For such a self-organized form of collective movement (not induced by a shared goal), which has been the focus of most past work, the only mechanism that we can think of is somehow introducing explicit local alignment into an egocentric model. While we have not implemented such an idea, it is not surprising for us to see that any model, even simple ones such as the Vicsek or Couzin model, gives rise to collective movement once you introduce local alignment as a model principle. However, collective movement and local alignment are mathematically intimately related and imply each other (thus, using one to explain the other does not provide much mechanistic insight), and the true challenge is to explain how local alignment emerges.

Reviewer N. 2:

5. Memory and Architectural Considerations

The current model functions essentially as a feedforward sensory-motor mapping, despite its recurrent attractor dynamics implementation. Adding even a brief discussion about how memory-based representations or hierarchical architectures might alter the observed behaviors would strengthen the paper’s connection to neuroscience. Contrasting "immediate embodied control" with "persistent internal models" would sharpen the conceptual positioning.

Response:

We thank the reviewer for this insightful comment. Using the neural field model introduced in the revision, we have discussed how memory-based cues can be integrated into our model via synaptic delay lines 698 to 702:

We note that the intensity of connections between neuron i and j (synaptic connectivity) can in general depend on time lag, $J_{ij}(t - t')$. Such a time-dependence can allow us to take pulse condition or synaptic delay into account [86]. While incorporating such a memory-based representation of neural dynamics may provide additional insights, here we do not investigate the consequences of synaptic delay.

Also, in the Supplementary Note.17 we present a conceptual generalization of our framework into a multilayer, potentially hierarchical network, based on Amari’s neural field model.

Furthermore, in the new extension of our model to account for coexistence via rapid switching, we introduce a continuous switching mechanism and discuss biological mechanisms, such as memory-based cues that can enable organisms to achieve such a continuous switch between allocentric and egocentric representations (lines 712 to 718).

Finally, we have added a section in the SI (referred to in Section 2.7) in which we have removed recurrent connections (by setting the synaptic connectivity equal to zero) and showed that such a reduction of the model does not give rise to collective movement, but rather a trivial phenomenology (also provided in a new SI video, SV.17). Thus, concluding that recurrent connections are essential for the rich phenomenology of the model.

Reviewer N. 2:

6. Reconciling Physical and Cognitive Modeling Approaches

While grounded in neurobiological elements (ring attractors in insects and fish), the model operates more like a physical system with tuned interactions than a cognitive or decision-making model as understood in neuroscience or AI. A clearer distinction between the physical modeling goals (dynamical systems, pattern formation) and cognitive interpretations (belief updating, decision-making) would help readers understand which level of biological explanation the model targets.

Response:

We thank the reviewer for this important observation. To address this comment we have provided, new models (as mentioned before), conceptual clarifications, and a better discussion of the literature. Below we list the steps taken to address this comment.

- We have developed a new model (the neural field model) and formulated the general conceptual framework (Supplementary Note. 17). The general formulation and the new model added in the revision are based on a classical biologically inspired approach (Amari’s neural field model) to model and simulate neural dynamics based on first principles (membrane potential). Thus the new model is better posed in the neuroscience and artificial intelligence literature. We believe this step has been essential in removing some of the limitations of the previous version, which was solely based on our spin system model, which, as the reviewer rightly noticed, leaned more toward physics tools and literature. This, not only makes the manuscript more connected with biology, neuroscience, and AI literature, but also provides important opportunities for the extension and exploitation of the framework in the future.

- To further clarify the biological and conceptual basis of our work and its goals, we have also added discussions of (Introduction, 2.1, and Methods)

- Mathematical and neuroscientific tools and concepts (neural field model and spin system models).
- The use of ring attractor networks in the context of “goal direction”.
- The empirical basis of allocentric and egocentric perceptions of space.
- As well as empirical background and our modeling goal.

We hope these extensive revisions remove some of the limitations of the previous version of the manuscript, significantly improve the manuscript, and better clarify the aim, scope, and level of biological explanation that our work aims to provide.

Reviewer N. 2:

Minor Suggestions

- The discussion of "criticality" follows convention, but claims about optimality near phase transitions should be presented carefully.

Response:

We have removed the term optimality or optimal from as we do not discuss optimality and as the reviewer notes, the use of this term was not careful. Rather, an “effective” decision-making region (in the sense of minimizing the measure of interest in the region studied) can be observed (qualitatively) close to (but not at) phase transitions.

Reviewer N. 2:

- References [54,55], used to suggest alignment is not fundamental, could be balanced with studies showing the opposite in other species for a more complete picture.

Response:

We have now balanced our discussion (Discussion lines 567 to 570) by explicitly citing key studies in fish and bird flocks that report high levels of alignment. We then emphasize that, despite these appearances, our allocentric model can produce the same flocking phenomenology without any built-in alignment rule, suggesting alignment in these taxa may itself be a downstream consequence of

allocentric sensory-motor processing rather than a primary interaction.

We note that many of the studies that argue for explicit alignment are based on the observed phenomenology, not behavioral rules. Thus, such studies are consistent with our model because our model gives rise to the same phenomenology (local alignment) as an emergent property, not a hard-wired rule (modeling assumption). On the other hand, some studies that have looked at this issue, as we have cited in the manuscript, have found no evidence for explicit local alignment to be a behavioral rule (lines 569 to 579):

While alignment is observed in many species—such as starlings [101, 102] and shoaling fish [38, 39]—our work suggests it may be an emergent by-product of allocentric (or coexistence of allocentric and egocentric) representation of space by animals— a view supported by studies that failed to find empirical evidence for explicit alignment among fish [38, 39] or swarm-forming locusts [40]. Our work shows that local alignment can arise as a form of consensus dynamic (not dissimilar, conceptually, to models of collective information acquisition [104]) for agents who have an allocentric representation of bearings (their own heading direction, and the bearings towards others, are within a world-centered frame). We show that, by contrast, if individuals exhibit an egocentric representation (whereby bearings are body-centered, but directional bearings are only encoded with reference to the present heading), social attraction can only result in the formation of relatively immobile aggregations. Here, the additive nature of attraction is analogous to “gravitational collapse” [105].

Reviewer N. 2:

- The figures, while informative, are quite dense. Consider simplifying captions or moving some exploratory visualizations to supplementary materials.

Response:

We have moved one of the figures to the supplementary information to make space for two new figures. We have also updated some of the figures and clarified captions to help readers understand the figures with more ease.

Reviewer N. 2:

Conclusion

This manuscript presents an elegant, well-executed theoretical model of collective behavior grounded in ring attractor dynamics. While it aims to bridge neuroscience and collective behavior, its approach leans toward physics-style modeling. With clearer terminology, better aligned citations, and more nuanced treatment of egocentric representations and belief modeling, this could make a valuable contribution to our understanding of collective behavior mechanisms.

Response:

We thank the reviewer once again for their positive assessment of our work. The extensive revision made in light of the reviewers’ constructive feedback has significantly improved the manuscript.

Response to reviewer

1 Overview of the changes

We would like to thank the editor and the referee once again for their constructive feedback.

Based on the constructive reviewer's suggestions, the Introduction is now shortened. In addition, minor revisions were made in two paragraphs in the Results, section 2.3 (lines 327 to 343), and a paragraph was added to the Discussion to better present the neural field model phenomenology and compare the results obtained with both model formulations. Finally, as suggested by the reviewer, we have added a paragraph in the Discussion in which we consider the functional aspects of our results. The manuscript is checked for accuracy.

Below, we provide our point-by-point responses to the reviewer's comments.

2 Response to the Reviewer

Reviewer:

I would like to thank the authors for their detailed response to my original review. To address the critiques raised in the first round of review, the authors made substantial changes to the paper.

They authors added a new model, i.e., neural field model, and analyzed and compared the prediction of this model with the original model based on a spin system. Furthermore, the authors extended their analysis to allow the switching between the allocentric and egocentric reference frameworks. The authors also make a few other improvements.

These additionally analysis substantially improved the paper, and addressed the major concerns in my initial review. I only have a few relatively minor concerns, mostly about the presentation of the paper.

Response:

We thank the reviewer, once again, for their valuable feedback, which helped us to substantially improve our manuscript.

Reviewer:

1. First, while the revised version of the Introduction made the modeling framework more grounded on the neuroscience literature, now it is very long now and the logic is a bit convoluted. I think the paper would benefit by streamline the Introduction.

Response:

We thank the reviewer for this valuable suggestion. We agree that the previous Introduction was too lengthy. We now provide a concise introduction. To maintain the grounding of the work in neuroscientific literature, which was added in the previous round in response to valuable feedback from the reviewer, we have moved the detailed discussion on neurobiological motivations to the beginning of the Result Section, 'Modeling framework', subheading, 'Neurobiological motivations', just before the models are introduced.

Reviewer:

Second, are there ways to empirically adjudicated between the two versions of the models (spin system and neural fields)? I feel the paper would benefit from a discussion on this issue. In addition, the similarities and differences in the predictions of the two models could be made more clear to better orient the readers

Response:

We thank the reviewer for this comment. To address this, we have added a new paragraph to the Discussion section that explicitly compares the predictions of the spin system and neural field model which can be used for testing of the empirical predictions of each model.

The new text in the discussion reads as follows (lines 599 to 618):

Despite the differences in their mathematical formulations, both the spin system and neural field models arrive at the same core prediction: an allocentric encoding of bearings is essential for the emergence of coordinated motion. However, they also exhibit differences that suggest avenues for empirical testing. For instance, the spin system model predicts random walk-like motion for high neural noise (low β), a pattern not observed in the neural field model. Conversely, the neural field model predicts speed changes in the presence of targets when β is small, with higher speeds occurring in response to more attractive stimuli. Furthermore, in the presence of a target, the egocentric neural field model can exhibit trajectory patterns distinct from those of the spin system model, such as spiral-like paths or slow movement away from an attractive stimulus when in very close proximity. Considering population-level properties, while the spin system model predicts there can be local, but not global, alignment of headings with an egocentric representation, no local order emerges in the neural field model. In addition, while similarities in collective patterns exist, our results suggest that milling behavior can be observed in some parameter ranges in the neural field model. We have not observed this pattern in the spin system model.

In addition, we have provided minor changes to the Results section, where individual motion in the neural field model is presented to better present and compare the phenomenology of the neural field model in comparison to the spin system model. The new text added in revision is indicated in blue (lines 328 to 343):

When considering tracking a moving or stationary target, we observe a generally similar phenomenology to the spin system model. Namely, the information acquisition capacity of the agent is maximized for intermediate values of β , where the agent exhibits more flexible decision-making. For too small values of β , the agent does not move or moves too slowly. For too large values of β , the agent lacks flexibility and performs poorly in finding or tracking a target. An allocentric agent performs better than an egocentric agent in tracking a moving target in the neural field model. In finding and staying close to a fixed target, allocentric and egocentric agents perform equally well for small β (this contrasts with the spin system model, in which egocentric agents perform better than allocentric ones for such values of β), but allocentric agents outperform an egocentric one for large β (similarly to the spin system model). See S.10 for details.

In the presence of a target, the neural field model also differs in several ways from the spin system model. For example, a stationary agent in the absence of a target can start to move (or a moving agent can increase its speed) towards the target, when the target is introduced. In addition, the threshold β value above which the agent starts to move is shifted to lower values in the presence of target(s) in the neural field model. Furthermore, in the egocentric neural field model (unlike the spin system model), we observe spiral-like trajectories during which the agent slowly moves towards, or away from, the target. See S.10 for details.

Reviewer:

Third, the authors show that the models exhibit different collective patterns under different parameter regimes. This work mainly comes from mechanistic considerations. Yet what might be the functional benefits of these different patterns? Do these considerations help us understand why certain patterns

are observed in certain scenarios for certain species? I think readers would be interested in hearing if there is any insight the authors may have. So a brief discussion on the functional aspects of the results might be useful.

Response:

We thank the reviewer for this insightful comment. To address this comment, we have added a paragraph in the discussion to clarify the implications of our work for a functional (and evolutionary) understanding of collective behavior. In this part, to avoid speculation, we have not explicitly discussed possible advantages of each pattern (as this was not a subject of our study and also, is rather poorly understood). Instead we have discussed this topic and cited relevant literature (5 new references). The added text reads as (618 to 633):

Here, we have shown that collective motion, along with a diverse set of patterns observed in animal groups, can arise directly from animals' navigational circuits. While this provides a mechanistic explanation, it is important to note that collective motion [106,107] and its distinct patterns [106]—such as fission–fusion dynamics [108,109], swirling/milling [110], and flash expansion [95–97]—may offer functional advantages and, therefore, be favored and shaped by evolution under different ecological contexts. In this regard, our model suggests that, although these patterns may also serve functional and evolutionary purposes, they can equally have proximate causes (rather than, or in addition to, evolutionary ones). Incorporating evolutionary perspectives into our framework—for example, by allowing our cognitive agents to evolve under different ecological scenarios—could help address how different motion patterns evolve, potentially by driving the system into specific parameter regimes and shaping the agents' sensory–motor integration in different ecological contexts. Such functional considerations could clarify how evolution shapes animal navigational circuits and their perception of the environment to better meet ecological and environmental demands. Ultimately, this approach may help explain—both mechanistically and functionally—why and how collective motion and its diversity emerge in a wide range of biological populations.

We would like to end by thanking the reviewer once again for their very helpful comments in both rounds of review.

Peer Review for "Allocentric Flocking" by Salahshour & Couzin

Summary

This paper introduces a compelling theoretical model of collective behavior based on ring attractor neural networks, comparing allocentric (world-centered) and egocentric (self-centered) spatial representations. The authors demonstrate that allocentric representations can generate coherent, large-scale flocking behaviors, while egocentric representations primarily produce static aggregations. Their model derives from Hamiltonian energy minimization principles and is explored through systematic simulations across various noise levels and social coupling strengths.

Strengths

- The paper is **technically sound** with clean model implementation and clear assumptions.
- It represents an excellent contribution to the field of **physics-inspired collective behavior models**.
- The simulations and phase space analysis are meticulously executed, with results that appear robust across different parameter regimes.
- The effort to ground flocking behavior in **neurobiological ring attractor mechanisms** provides a fresh and intriguing perspective.

Major Comments

1. Neuroscience Terminology and Framing

The paper uses several terms from cognitive neuroscience in ways that might not align with their conventional meanings:

- "Belief updating" typically suggests probabilistic or Bayesian inference processes, whereas here it refers to deterministic shifts in attractor network activity. I'd recommend clarifying this usage or selecting alternative terminology.
- Similarly, "internal model" and "world model" appear without distinguishing between basic embodied representations (as found in insects) and sophisticated predictive models in complex organisms. This distinction would strengthen the paper's theoretical foundations.
- The supporting references for "internal models" (e.g., [22-25]) could be better aligned with the established neuroscience literature.

2. Suggested additions for consideration:

- For neural "world models": Keller & Mrsic-Flogel (2018, *Neuron*), Pezzulo et al. (2017, *TICS*)
- For computational world models: Ha & Schmidhuber (2018), Lehman et al. (2022)

3. Model Construction and Emergent Properties

While elegant, the model seems somewhat constructed to yield its primary finding—that allocentric representations produce stable flocking while egocentric ones don't. The ring attractor with energy minimization and specified input fields makes this outcome somewhat expected.

I would appreciate a more explicit acknowledgment that the observed behaviors are substantially encoded in the system's structure rather than emerging unexpectedly. This doesn't diminish the work's value but would position it more accurately within the field.

A discussion comparing this framework to other Ising-like models in neuroscience would help highlight

what's genuinely novel about the navigational coupling introduced here.

4. Treatment of Egocentric Representations

The manuscript's conclusion that egocentric representation fails to support collective motion risks overgeneralization. In biological systems, egocentric representations work alongside velocity integration, memory, and goal-directed behaviors—elements absent from this model.

I'd be interested in the authors' thoughts on what mechanisms might enable egocentric flocking in real systems, such as short-term memory, learned movement patterns, or heading stabilization.

5. Memory and Architectural Considerations

The current model functions essentially as a feedforward sensory-motor mapping, despite its recurrent attractor dynamics implementation. Adding even a brief discussion about how memory-based representations or hierarchical architectures might alter the observed behaviors would strengthen the paper's connection to neuroscience.

Contrasting "immediate embodied control" with "persistent internal models" would sharpen the conceptual positioning.

6. Reconciling Physical and Cognitive Modeling Approaches

While grounded in neurobiological elements (ring attractors in insects and fish), the model operates more like a physical system with tuned interactions than a cognitive or decision-making model as understood in neuroscience or AI.

A clearer distinction between the physical modeling goals (dynamical systems, pattern formation) and cognitive interpretations (belief updating, decision-making) would help readers understand which level of biological explanation the model targets.

Minor Suggestions

- The discussion of "criticality" follows convention, but claims about optimality near phase transitions should be presented carefully.
- References [54,55], used to suggest alignment is not fundamental, could be balanced with studies showing the opposite in other species for a more complete picture.
- The figures, while informative, are quite dense. Consider simplifying captions or moving some exploratory visualizations to supplementary materials.

Conclusion

This manuscript presents an elegant, well-executed theoretical model of collective behavior grounded in ring attractor dynamics. While it aims to bridge neuroscience and collective behavior, its approach leans toward physics-style modeling. With clearer terminology, better aligned citations, and more nuanced treatment of egocentric representations and belief modeling, this could make a valuable contribution to our understanding of collective behavior mechanisms.